# Dynamic Prompt Learning: Addressing Cross-Attention Leakage for Text-Based Image Editing

**Kai Wang**[1,2], **Fei Yang**[3,*] **Shiqi Yang**[1,2], **Muhammad Atif Butt**[1,2], **Joost van de Weijer**[1,2]

[1]Computer Vision Center, Barcelona, Spain
[2] Universitat Autonoma de Barcelona, Barcelona, Spain
[3] College of Computer Science, Nankai University, Tianjin, China
`{kwang,syang,mabutt,joost}@cvc.uab.es, feiyang@nankai.edu.cn`

## Abstract

Large-scale text-to-image generative models have been a ground-breaking development in generative AI, with diffusion models showing their astounding ability to synthesize convincing images following an input text prompt. The goal of image editing research is to give users control over the generated images by modifying the text prompt. Current image editing techniques are susceptible to unintended modifications of regions outside the targeted area, such as on the background or on distractor objects which have some semantic or visual relationship with the targeted object. According to our experimental findings, inaccurate cross-attention maps are at the root of this problem. Based on this observation, we propose *Dynamic Prompt Learning* (*DPL*) to force cross-attention maps to focus on correct *noun* words in the text prompt. By updating the dynamic tokens for nouns in the textual input with the proposed leakage repairment losses, we achieve fine-grained image editing over particular objects while preventing undesired changes to other image regions. Our method *DPL*, based on the publicly available *Stable Diffusion*, is extensively evaluated on a wide range of images, and consistently obtains superior results both quantitatively (CLIP score, Structure-Dist) and qualitatively (on user-evaluation). We show improved prompt editing results for Word-Swap, Prompt Refinement, and Attention Re-weighting, especially for complex multi-object scenes.

## 1 Introduction

Text-to-Image (T2I) is advancing at a revolutionary pace and has demonstrated an unprecedented ability to generate diverse and realistic images [27, 33, 34, 35]. The state-of-the-art T2I models are trained on extremely large language-image datasets, which require huge computational resources. However, these models do not allow for straightforward *text-guided image editing* and generally lack the capability to control specific regions of a given image.

Recent research on *text-guided image editing* allows users to easily manipulate an image using only text prompts [5, 9, 14, 15, 24, 42, 43]. In this paper, we focus on prompt based image editing, where we aim to change the visual appearance of a target object (or objects) in the image. Several of the existing methods use DDIM inversion [40] to attain the initial latent code of the image and then apply their proposed editing techniques along the denoising phase [28, 30, 42]. Nevertheless, current text-guided editing methods are susceptible to unintentional modifications of image areas. We distinguish between two major failure cases: unintended changes of background regions and unintended changes of distractor objects (objects that are semantically or visually related to the target object), as shown in Fig. 1. The functioning of existing editing methods greatly depends on

---

*Corresponding author

37th Conference on Neural Information Processing Systems (NeurIPS 2023).

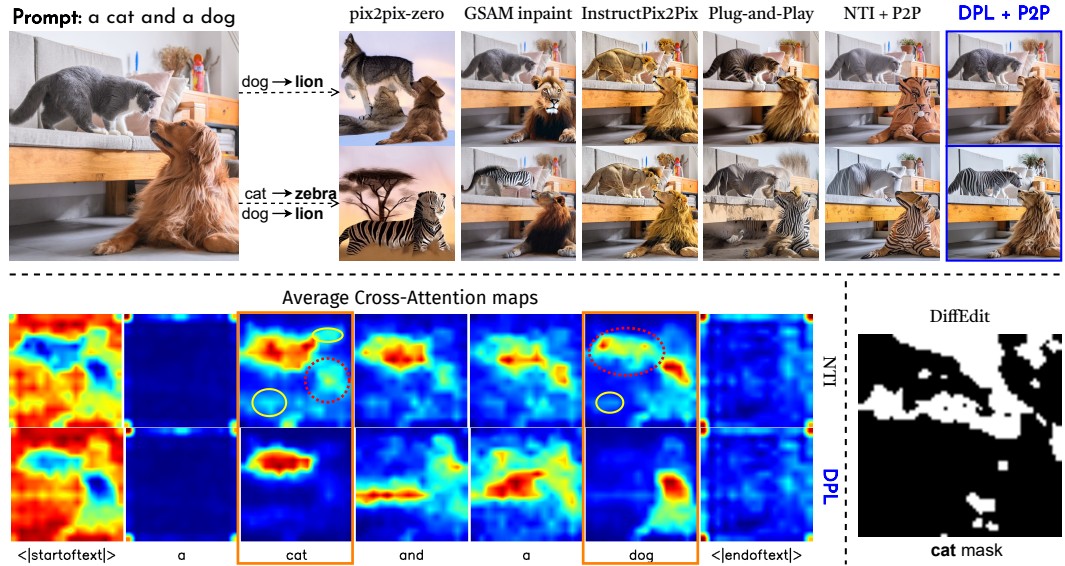

Figure 1: Compared with other image editing methods, our *DPL* achieves more consistent layout when modifying one or more objects in the image and keeping other content frozen. And also the desired editing is correctly delivered to the corresponding area. DiffEdit [9] method thoroughly fails to detect the editing region in multi-object cases. And Null-Text inversion (NTI [28]) is unable to perfectly distinguish the objects in the given image since the cross-attention maps are suffering background leakage (yellow circle) and distractor object leakage (red circle). *DPL* can more successfully localize the objects in the textual prompts, thus benefits for future editing. Here the $16 \times 16$ cross-attention maps are *interpolated* to the image size for better view. (image credits: gettyimages)

the accuracy of the cross-attention maps. When visualizing the cross-attention maps (see Fig. 1), we can clearly observe that for DDIM [40] and Null-Text inversion [28] these maps do not only correlate with the corresponding objects. We attribute this phenomenon to *cross-attention leakage* to background and distractor objects. Cross-attention leakage is the main factor to impede these image editing methods to work for complex backgrounds and multi-object images. Similar observations for failure cases are also shown in [9, 14, 19, 22, 28]. For example, Custom Diffusion [22] fails for *multi-concept composition* and DiffEdit does not correctly detect the mask region for a given concept.

We conduct a comprehensive analysis of the Stable Diffusion (SD) model and find that the inaccurate attribution of cross-attention (resulting in undesirable changes to background and distractor objects) can be mitigated by exploiting the semantic clusters that automatically emerge from the self-attention maps in the diffusion model. Instead of fine-tuning the diffusion model, which potentially leads to *catastrophic neglect*[7], we propose replacing prompts related to scene objects with dynamic prompts that are optimized. We propose two main losses for *dynamic prompt learning* (DPL), one tailored to reduce attention leakage to distractor objects and one designed to prevent background attention leakage. Moreover, better cross-attention maps are obtained when we update the prompts for each timestamp (as implied by *dynamic*). Furthermore, to facilitate meaningful image editing, we use Null-Text Inversion [28] with classifier-free guidance [17]. In the experiments, we quantitatively evaluate the performance of *DPL* on images from LAION-5B [39] dataset. And *DPL* shows superior evaluation metrics (including CLIP-Score and Structure-Dist) and improved user evaluation. Furthermore, we show prompt-editing results for Word-Swap, Prompt Refinement, and Attention Re-weighting on complex multi-object scenes.

## 2 Related work

For text-guided image editing with diffusion models, various methods [2, 8, 20, 23, 24] leverage CLIP [32] for image editing based on a pretrained *unconditional* diffusion model. However, they are limited to the generative prior which is only learned from visual data of a specific domain, whereas the CLIP text-image alignment information is from much broader data. This prior knowledge gap is

mitigated by recent progress of text-to-image (T2I) models [6, 10, 18, 33, 34, 38]. Nevertheless, these T2I models offer little control over the generated contents. This creates a great interest in developing methods to adopt such T2I models for controllable image editing. SDEdit [26], a recent work utilizing diffusion models for image editing, follows a two-step process. It introduces noise to the input image and then applies denoising using the SDE prior to enhance realism and align with user guidance. Imagic [19] and P2P [15] attempt structure-preserving editing via Stable Diffusion (SD) models. However, Imagic [19] requires fine-tuning the entire model for each image and focuses on generating variations for the objects. P2P [15] has no need to fine-tune the model, it retrains the image structure by assigning *cross-attention* maps from the original image to the edited one in the corresponding text token. InstructPix2Pix [4] is an extension of P2P by allowing human-like instructions for image editing. To make the P2P capable of handling real images, Null-Text inversion (NTI) [28] proposed to update the *null-text* embeddings for accurate reconstruction to accommodate with the classifier-free guidance [17]. Recently, pix2pix-zero [30] propose noise regularization and *cross-attention* guidance to retrain the structure of a given image. However, it only supports image translation from one concept to another one and is not able to deal with multi-concept cases. DiffEdit [9] automatically generates a mask highlighting regions of the input image by contrasting predictions conditioned on different text prompts. Plug-and-Play (PnP) [42] demonstrated that the image structure can be preserved by manipulating spatial *features* and *self-attention* maps in the T2I models. MnM [31] clusters self-attention maps and compute their similarity to the cross attention maps to determine the object localization, thus benefits to generate object-level shape variations. StyleDiffusion [24] adds up a mapping network to modify the value component in cross-attention computation for regularizing the attentions and utilizes P2P for image editing.

These existing methods highly rely on precise *cross-attention* maps and overlook an important case where the cross-attention maps may not be perfectly connected to the concepts given in the text prompt, especially if there are multiple objects in the given image. Therefore, these imperfect cross-attention maps may lead to undesirable editing leakage to unintended regions. Essentially, our method *DPL* complements existing methods to have more semantic richer cross-attention maps. There are various diffusion-based image editing methods relying on giving a mask manually [29, 1] or detecting a mask automatically (GSAM-inpaint [44]) using a pretrained segmentation model. On the contrary, our method only requires textual input and automatically detects the spatial information. This offers a more intuitive editing demand by only manipulating the text prompts.

There are some papers [21, 12] concentrated on object removal or inpainting using user-defined masks [1, 29, 44]. In contrast, the act of placing an object in a suitable position without relying on a mask is an active research domain. This direction usually entails fine-tuning models for optimal outcomes and runs parallel to the research direction explored in this paper.

In this paper, we are also related to the transfer learning for T2I models [22, 37, 11, 13, 25], which aims at adapting a given model to a *new concept* by given images from the users and bind the new concept with a unique token. However, instead of finetuning the T2I model [25, 13, 37], we learn new concept tokens by personalization in the text embedding space, which is proposed in Textual Inversion [11]. By this means, we avoid updating a T2I model for each concept separately. Furthermore, we step forward to the scenario with complex background or multiple objects, where the previous methods have not considered. In contrast to transfer learning approaches, which aim to ensure that the learned concepts can accurately generate corresponding objects in given images, our proposed approach *DPL* focuses on enhancing image editing tasks by regularizing cross-attention to disentangle their positions.

## 3 Method

In this section, we first provide a short preliminary and then describe our method to achieve this demand. An illustration of our method is shown in Fig. 2 and Algorithm 1.

### 3.1 Preliminary

**Latent Diffusion Models.** We use Stable Diffusion v1.4 which is a Latent Diffusion Model (LDM) [35]. The model is composed of two main components: an autoencoder and a diffusion model. The encoder $\mathcal{E}$ from the autoencoder component of the LDMs maps an image $\mathcal{I}$ into a latent code $z_0 = \mathcal{E}(\mathcal{I})$ and the decoder reverses the latent code back to the original image as $\mathcal{D}(\mathcal{E}(\mathcal{I})) \approx \mathcal{I}$. The

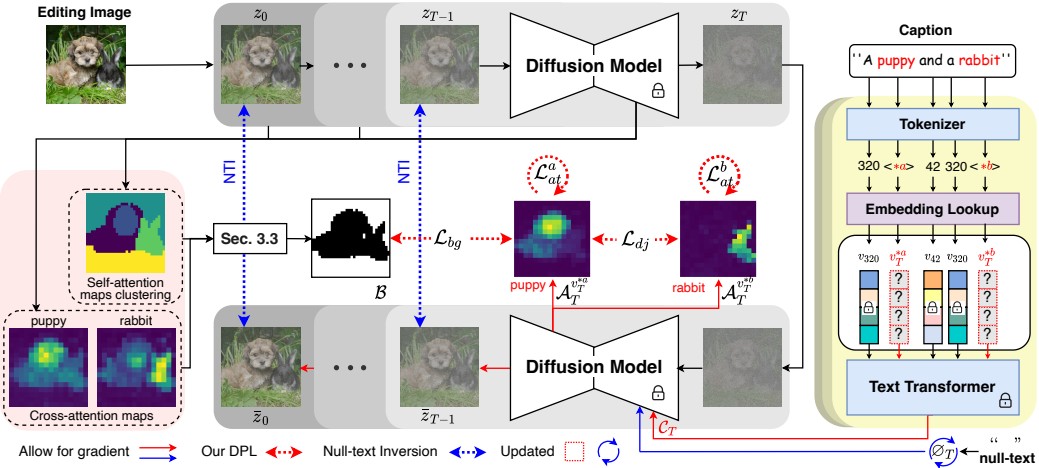

Figure 2: Dynamic Prompt Learning (*DPL*) first transforms the noun words in the text prompt into dynamic tokens. We register them in the token dictionary and initialize the representations by their original words at time $T$. Using DDIM inversion, we condition the DM-UNet with the original text condition $\mathcal{C}$ to get the diffusion trajectory $\{z_t\}_0^T$ and background mask $\mathcal{B}$. In each denoising step $\bar{z}_{t-1} \rightarrow \bar{z}_t$, we first update the dynamic token set $\mathcal{V}_t$ with a background leakage loss, a disjoint object attention loss and an attention balancing loss, in order to ensure high-quality cross-attention maps. Then we apply Null-Text inversion to approximate the diffusion trajectory.

diffusion model can be conditioned on class labels, segmentation masks or textual input. Let $\tau_\theta(y)$ be the conditioning mechanism which maps a condition $y$ into a conditional vector for LDMs, the LDM model is updated by the loss:

$$L_{LDM} := \mathbb{E}_{z_0 \sim \mathcal{E}(x), y, \epsilon \sim \mathcal{N}(0,1), t \sim \text{Uniform}(1,T)} \left[ \|\epsilon - \epsilon_\theta(z_t, t, \tau_\theta(y))\|_2^2 \right] \tag{1}$$

The neural backbone $\epsilon_\theta$ is typically a conditional UNet [36] which predicts the added noise. More specifically, text-guided diffusion models aim to generate an image from a random noise $z_T$ and a conditional input prompt $\mathcal{P}$. To distinguish from the general conditional notation in LDMs, we itemize the textual condition as $\mathcal{C} = \tau_\theta(\mathcal{P})$.

**DDIM inversion.** Inversion entails finding an initial noise $z_T$ that reconstructs the input latent code $z_0$ upon sampling. Since we aim at precisely reconstructing a given image for future editing, we employ the deterministic DDIM sampling [40]:

$$z_{t+1} = \sqrt{\bar{\alpha}_{t+1}} f_\theta(z_t, t, \mathcal{C}) + \sqrt{1 - \bar{\alpha}_{t+1}} \epsilon_\theta(z_t, t, \mathcal{C}) \tag{2}$$

where $\bar{\alpha}_{t+1}$ is noise scaling factor defined in DDIM [40] and $f_\theta(z_t, t, \mathcal{C})$ predicts the final denoised latent code $z_0$ as $f_\theta(z_t, t, \mathcal{C}) = \left[ z_t - \sqrt{1 - \bar{\alpha}_t} \epsilon_\theta(z_t, t, \mathcal{C}) \right] / \sqrt{\bar{\alpha}_t}$.

**Null-Text inversion.** To amplify the effect of conditional textual input, classifier-free guidance [17] is proposed to extrapolate the conditioned noise prediction with an unconditional noise prediction. Let $\varnothing = \tau_\theta(\text{``''})$ denote the null text embedding, then the classifier-free guidance is defined as:

$$\tilde{\epsilon}_\theta(z_t, t, \mathcal{C}, \varnothing) = w \cdot \epsilon_\theta(z_t, t, \mathcal{C}) + (1 - w) \cdot \epsilon_\theta(z_t, t, \varnothing) \tag{3}$$

where we set the guidance scale $w = 7.5$ as is standard for Stable Diffusion [17, 28, 35]. However, the introduction of amplifier-free guidance complicates the inversion process, and the generated image based on the found initial noise $z_T$ deviates from the original image. Null-text inversion [28] proposes a novel optimization which updates the null text embedding $\varnothing_t$ for each DDIM step $t \in [1, T]$ to approximate the DDIM trajectory $\{z_t\}_0^T$ according to:

$$\min_{\varnothing_t} \|\bar{z}_{t-1} - \tilde{\epsilon}_\theta(\bar{z}_t, t, \mathcal{C}, \varnothing_t)\|_2^2 \tag{4}$$

where $\{\bar{z}_t\}_0^T$ is the backward trace from Null-Text inversion. Finally, this allows to edit real images starting from initial noise $\bar{z}_T = z_T$ using the learned null-text $\varnothing_t$ in combination with P2P [15].

Note, that the cross-attention maps of Null-Text inversion ($w = 7.5$) and DDIM inversion ($w = 1.0$) are similar, since NTI is aiming to approximate the same denoising path.

**Prompt2Prompt.** P2P [15] proposed to achieve editing by manipulating the cross-attention $\mathcal{A}_t^*$ from target input $\mathcal{P}^*$ according to the cross-attention $\mathcal{A}_t$ from the source input $\mathcal{P}$ in each timestamp $t$. The quality of the estimated attention maps $\mathcal{A}_t$ is crucial for the good functioning of P2P. This paper addresses the problem of improved $\mathcal{A}_t$ estimation for complex images.

### 3.2   *DPL*: Dynamic Prompt Learning for addressing cross-attention leakage

Prompt-based image editing takes an image $\mathcal{I}$ described by an initial prompt $\mathcal{P}$, and aims to modify it according to an altered prompt $\mathcal{P}^*$ in which the user indicates desired changes. The initial prompt is used to compute the cross-attention maps given the conditional text input $\mathcal{P}$. As discussed in Sec. 1, the quality of cross-attention maps plays a crucial role for prompt-based image editing methods (see Fig. 1), since these attention maps will be used during the generation process with the altered user prompt [15]. We identify two main failures in the attention map estimation: **(1)** distractor object leakage, which refers to wrong assignment of attention to a semantically or visually similar foreground object and **(2)** background leakage, referring to undesired attention on the background. Both of these problems result in artifacts or undesired modifications in generated images.

The cross-attention maps in the Diffusion Model UNet are obtained from $\epsilon_\theta(z_t, t, \mathcal{C})$, which is the first component in Eq. 3. They are computed from the deep features of noisy image $\psi(z_t)$ which are projected to a query matrix $Q_t = l_Q(\psi(z_t))$, and the textual embedding which is projected to a key matrix $K = l_K(\mathcal{C})$. Then the attention map is computed according to:

$$\mathcal{A}_t = softmax(Q_t \cdot K^T / \sqrt{d}) \tag{5}$$

where $d$ is the latent dimension, and the cell $[\mathcal{A}_t]_{ij}$ defines the weight of the $j$-th token on the pixel $i$.

In this paper, we propose to optimize the word embeddings $v$ corresponding to the initial prompt $\mathcal{P}$, in such a way that the resulting cross-attention maps $\mathcal{A}_t$ do not suffer from the above-mentioned problems, *i.e.* distractor-object and background leakage. We found that optimizing the embedding $v_t$ separately for each timestamp $t$ leads to better results than merely optimizing a single $v$ for all timestamps. The initial prompt $\mathcal{P}$ contains $K$ noun words and their corresponding learnable tokens at each timestamp $\mathcal{V}_t = \{v_t^1, ..., v_t^k ..., v_t^K\}$. We update each specified word in $\mathcal{V}_t$ for each step $t$.

The final sentence embedding $\mathcal{C}_t$ now varies for each timestamp $t$ and is computed by applying the text transformer on the text embeddings: some of which are learned (those referring to nouns) while the rest remains constant for each timestamp (see Fig. 2). For example, given a prompt "a puppy and a rabbit" and the updatable noun set {puppy, rabbit}, in each timestamp $t$ we have transformed the textual input into trainable ones as "a <puppy> and a <rabbit>" and we aim to learn word embeddings $v_t^{\text{<puppy>}}$ and $v_t^{\text{<rabbit>}}$ in each timestamp $t$.

We propose several losses to optimize the embedding vectors $\mathcal{V}_t$: we develop one loss to address cross-attention leakage due to object distractors, and another one to address background leakage. We also found that the attention balancing loss, proposed by Attend-and-Excite [7], was useful to ensure each specified noun word from $\mathcal{P}$ is focusing on its relevant concept.

**Disjoint Object Attention Loss.** To address the problem of distractor object attention leakage, we propose a loss that minimizes the attention overlap between the attention maps corresponding to the different objects, that is an updatable noun set $\mathcal{V}_t$. This prevents specific objects related regions to be linked to multiple nouns, which is one of the sources of distractor object attention. We propose to minimize a cosine similarity loss among the $K$ cross-attention maps (referring to the noun words in the input prompt $\mathcal{P}$) to avoid their overlapping in cross-attention maps through:

$$\mathcal{L}_{dj} = \sum_{i=1}^{K} \sum_{\substack{j=1 \\ i \neq j}}^{K} \cos(\mathcal{A}_t^{v_t^i}, \mathcal{A}_t^{v_t^j}) \tag{6}$$

**Background Leakage Loss.** In order to alleviate the background leakage issue, we propose to utilize background masks to keep the background area unchanged. Details on how to get the background mask are discussed in Sec. 3.3. After obtaining the BG mask $\mathcal{B}$ of the given image, for each $v_t^k \in \mathcal{V}_t$,

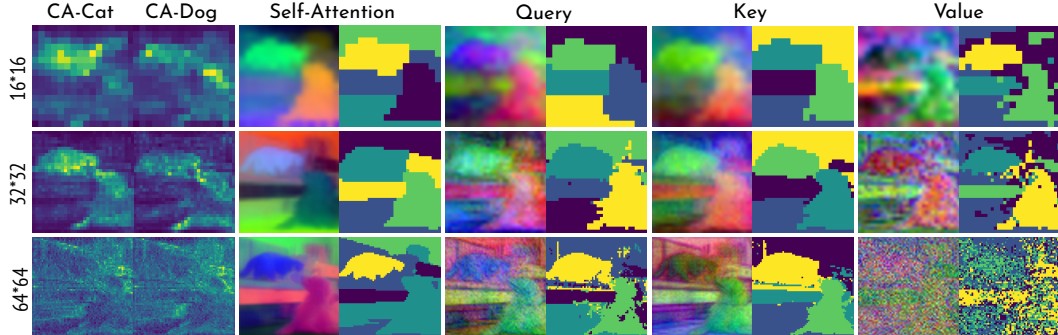

Figure 3: After DDIM inversion, we visualize the average of each component over $T$ timestamps for the input image in Fig. 1 (with the prompt: "a cat and a dog"). From left to right, they are cross-attention (CA) for each *noun* word, then PCA&clustering of the self-attention (SA), query, key and value representations. Here, the maps are simply *resized* without any interpolation to demonstrate their original views.

we force its cross-attention map to have minimal overlap with the background region:

$$\mathcal{L}_{bg} = \sum_{i=1}^{K} \cos(\mathcal{A}_t^{v_t^i}, \mathcal{B}) \tag{7}$$

**Attention Balancing Loss.** For each $v_t^k \in \mathcal{V}_t$, we pass its cross-attention map through a Gaussian filter $\mathcal{F}$ and formulate a set of losses $\mathcal{L}_{v_t^k}$ for all words in $\mathcal{V}_t$. The attention loss is defined as the maximum value in $\mathcal{L}_{v_t^k}$:

$$\mathcal{L}_{at} = \max_{v_t^k \in \mathcal{V}_t} \mathcal{L}_{v_t^k} \qquad \text{where} \qquad \mathcal{L}_{v_t^k} = 1 - \max[\mathcal{F}(\mathcal{A}_t^{v_t^k})] \tag{8}$$

The attention loss will strengthen the cross-attention activations for *localizing* token $v_t^k$, thus force $\mathcal{A}_t^v$ to not leak attentions to other irrelevant concepts in the image. In other words, the resulting cross-attention maps will be more concentrated on the corresponding object region. Different from Attend-and-Excite [7] where the refinement loss is applied to update the latent code to augment the multiple object generation, we are the first to explore its role in learning new conceptual tokens.

With all these three losses, we aim to update the learnable token $v_t^k$ as:

$$\arg\min_{\mathcal{V}_t} \mathcal{L} \quad \text{where} \quad \mathcal{L} = \lambda_{at} \cdot \mathcal{L}_{at} + \lambda_{dj} \cdot \mathcal{L}_{dj} + \lambda_{bg} \cdot \mathcal{L}_{bg} \tag{9}$$

Note that, when there is only one updatable noun token in the text input $\mathcal{P}$, we only apply the $\mathcal{L}_{bg}$ to relieve the background leakage since other losses rely on at least two words. In this paper, we apply *DPL* over the $16^2$ attention following the common practices [15, 7] for a fair comparison.

**Gradual Optimization for Token Updates.** So far, we introduced the losses to learn new dynamic tokens at each timestamp. However, the cross-attention leakage is gradually accumulated in the denoising phase. Hence, we enforce all losses to reach a pre-defined threshold at each timestamp $t$ to avoid overfitting of the cross-attention maps. We express the gradual threshold by an exponential function. For the losses proposed above, the corresponding thresholds at time $t$ are defined as:

$$TH_t^{at} = \beta_{at} \cdot \exp(-t/\alpha_{at}); \quad TH_t^{dj} = \beta_{dj} \cdot \exp(-t/\alpha_{dj}); \quad TH_t^{bg} = \beta_{bg} \cdot \exp(-t/\alpha_{bg}). \tag{10}$$

We verify the effectiveness of this mechanism together with other hyperparameters in our ablation experiments (see Fig. 5-(a) and Supplementary Material.).

**Null-Text embeddings.** The above described token updating ensures that the cross-attention maps are highly related to the noun words in the text prompt and minimize cross-attention leakage. To

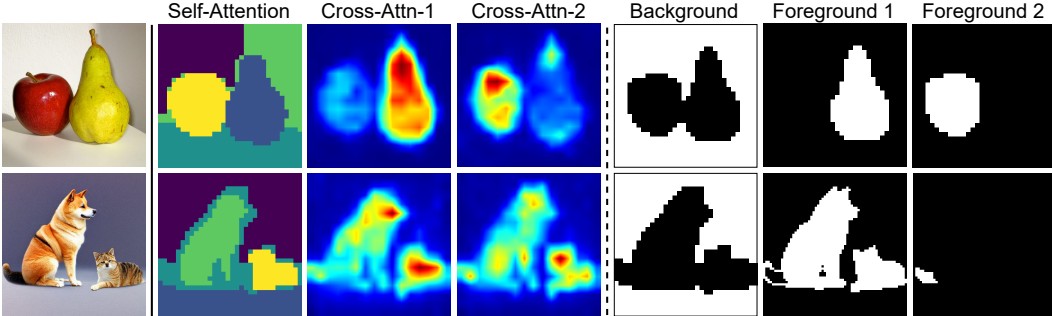

| | Self-Attention | Cross-Attn-1 | Cross-Attn-2 | Background | Foreground 1 | Foreground 2 |

Figure 4: Although cross-attention maps produce acceptable results in certain cases (up) with minor leakage, the object masks from self-cross attention matching are not always accurate due to the leakage problem (down). However, as the cross-attention maps tend to concentrate on the foreground objects, the background masks remain reliable.

be able to reconstruct the original image, we use Null-Text inversion [28] in addition to learn a set of null embeddings $\varnothing_t$ for each timestamp $t$. Then at the time $t$, we have a set of learnable word embeddings $\mathcal{V}_t$ and null text $\varnothing_t$ which can accurately localize the objects and also reconstruct the original image.

### 3.3 Attention-based background estimation

To address the background attention leakage, we investigate the localization information of self-attention and feature representations, which are present in the diffusion model. Besides cross-attention (CA), during the inversion of an image with the DDIM sampler, we can also obtain self-attention maps (SA), or directly analyze the query, key and value in the self-attention maps. In Fig. 3 we visualize these maps. The main observation is that self-attention maps more closely resemble the semantic layout. We therefore exploit these to reduce background leakage.

To acquire the background via self-cross attention matching, we use the agreement score [31] between cross-attention $\mathcal{A}^n$ of noun words $n$ and each of the self-attention clusters $\mathcal{M}_v, v \in [1, V]$, defined as $\sigma_{nv} = \sum(\mathcal{A}^n \cdot \mathcal{M}_v)/\sum \mathcal{M}_v$.

However, it should be noted that the agreement score does not accurately segment the objects, since there is still attention leakage between objects and distractor objects (see Fig. 4 for examples). Therefore, other than localizing objects as in [31], we use the agreement score only for background mask estimation. We label the cluster $\mathcal{M}_v$ as being background (BG) if for all $n$

**Algorithm 1:** Dynamic Prompt Learning

1 **Input:** A source prompt $\mathcal{P}$, an input image $\mathcal{I}$
2 **Output:** A noise vector $\bar{z}_T$, a set of updated tokens $\{\mathcal{V}_t\}_1^T$ and null-text embeddings $\{\varnothing_t\}_1^T$
3 Set $T = 50$ and scale $w = 1$;
4 $\{z_t\}_0^T, \mathcal{B} \leftarrow$ DDIM-inv($\mathcal{I}$);
5 Set guidance scale $w = 7.5$;
6 Initialize $\mathcal{V}_T$ with original noun tokens;
7 Initialize $\bar{z}_T = z_T, \mathcal{P}_T = \mathcal{P}, \varnothing_T = \tau_\theta(\text{""})$;
8 **for** $t = T, T-1, \ldots, 1$ **do**
9     Initialize $\mathcal{P}_t$ by $\mathcal{V}_t$, then $\mathcal{C}_t = \tau_\theta(\mathcal{P}_t)$;
10     **while** $\mathcal{L}_{at} \geq TH_t^{at}$ *or* $\mathcal{L}_{dj} \geq TH_t^{dj}$ *or* $\mathcal{L}_{bg} \geq TH_t^{bg}$ **do**
11         $\mathcal{L} = \lambda_{at} \cdot \mathcal{L}_{at} + \lambda_{dj} \cdot \mathcal{L}_{dj} + \lambda_{bg} \cdot \mathcal{L}_{bg}$
12         $\mathcal{V}_t \leftarrow \mathcal{V}_t - \nabla_{\mathcal{V}_t}\mathcal{L}$
13     **end**
14     Update $\mathcal{P}_t$ by $\mathcal{V}_t$, then $\mathcal{C}_t = \tau_\theta(\mathcal{P}_t)$;
15     $\tilde{z}_t = \tilde{\epsilon}_\theta(\bar{z}_t, t, \mathcal{C}_t, \varnothing_t)$
16     $\bar{z}_{t-1}, \varnothing_t \leftarrow NTI(\tilde{z}_t, \varnothing_t)$
17     Initialize $\varnothing_{t-1} = \varnothing_t, \mathcal{V}_{t-1} = \mathcal{V}_t$
18 **end**
19 **Return** $\bar{z}_T, \{\mathcal{V}_t\}_1^T, \{\varnothing_t\}_1^T$

if $\sigma_{nv} < TH$, where $TH$ is a threshold. Empirically, we choose self-attention with size 32, cross-attention with size 16 and $TH = 0.2, V = 5$ to obtain the background mask $\mathcal{B}$.

## 4 Experiments

We demonstrate our method in various experiments based on the open-source T2I model Stable Diffusion [35] following previous methods [42, 30, 28]. All experiments are done on an A40 GPU.

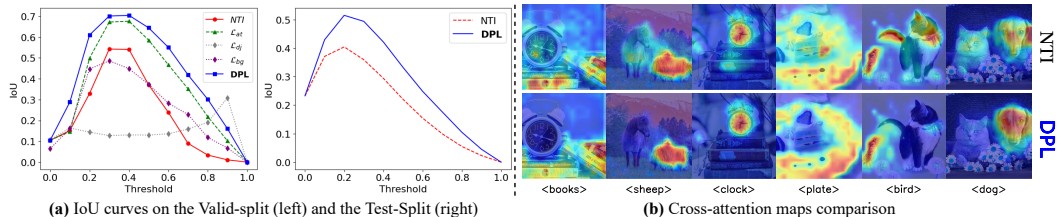

**(a)** IoU curves on the Valid-split (left) and the Test-Split (right)

**(b)** Cross-attention maps comparison

Figure 5: (a) IoU curves drawn by varying the threshold from 0.0 to 1.0 for *MO-Set*. (b) We show some images for comparisons between the Null-Text inversion [28] and our method *DPL* demonstrated in cross-attention maps.

**Dataset.** We focus here on real-image editing, but also provide examples of generated image editing in the Supplementary Material. We use clip retrieval [3] to obtain experimental multi-object real images from the LAION-5B [39] dataset by matching CLIP embeddings of the source text description. For the ablation study, we select 100 multi-object images from various search prompts (including concepts as bird, dog, cat, book, clock, pen, pizza, table, horse, sheep, etc.). This Multi-Object set is abbreviated with *MO-Set*. Of the images, 40 images are used for validation and the other 60 are test set. We manually verify their segmentation maps which serve as groundtruth in the ablation study IoU computation. Finally, for user study, we create a *URS-Set* of 60 multi-object images of semantically related objects from various concepts.

**Comparison methods.** For qualitative comparison, we compare with pix2pix-zero [30], GSAM-inpaint [44], InstructPix2Pix [4], Plug-and-Play [42] and NTI+P2P [28]. For quantitative evaluation, we benchmark our approach against NTI+P2P, considering it as a robust baseline for maintaining content stability and achieving superior conceptual transfer.

## 4.1 Ablation study

The ablation study is conducted on the *MO-Set* set. To quantitatively evaluate the performance of *DPL* in localization, we vary the threshold from 0.0 to 1.0 to obtain the segmentation mask from the cross-attention maps and compute the IoU metric using the segmentation groundtruth for comparison.

In Fig. 5-(a), we compare the localizing performance with Null-Text inversion (NTI), *DPL* with each proposed loss separately and our final model. We can observe that the disjoint object attention loss and the background leakage loss are not able to work independently to improve the cross-attention map. The attention balancing loss can enhance the cross-attention quality and can be further boosted by the other two proposed losses. We use the validation set to optimize the various hyperparameters (see Eq. 9 and Eq. 10). Using these parameters, we show similar results on the independent test-set (Fig. 5-(a)). These parameters are set for all further results (more details are provided in Supplementary). In Fig. 5-(b), we qualitatively verify our method on some examples of comparison with cross-attention maps. *DPL* improves the cross-attention quality than NTI as the baseline method.

## 4.2 Image editing evaluation

For image editing, we combine our proposed *DPL* with the P2P [15] image editing method. Here we mainly focus on Word-Swap but also include more results for Prompt Refinement, and Attention Re-weighting in the supplementary material.

**Word-Swap.** In the word swapping scenario, we quantitatively compare NTI and *DPL* under two scenarios shown in Table 1. For each case, we use the collected 20 images with searching prompts "a cat and a dog" and "a box and a clock" to estimate the evaluation metrics, including CLIP-Score [16] and Structure Dist [41]. In both cases, *DPL* is getting better performance than the NTI baseline (more shown in Supplementary).

Furthermore, we conduct a forced-choice user study involving 20 evaluators and the assessment of 60 image editing pairs. Users were asked to evaluate 'which result does better represent the prompt change?' (more protocol choices can be found in Supplementary). We can observe a significantly higher level of satisfaction among the participants with our method when compared to NTI.

Table 1: Comparison between our method *DPL* +P2P and the baseline NTI+P2P by modifying multiple objects simultaneously. we examined the quality of editing by transforming an image depicting "a cat and a dog" into "a leopard and a tiger". We evaluated the editing quality of both approaches using objective metrics, such as CLIP-Score and Structure-Dist, across these multi-object collections. Additionally, to capture the subjective opinions of users, we conducted a user study involving 60 pairs of multi-object image translations.

| Method | cat → leopard and dog → tiger | | book → box and clock → sunflower | | User Study |
|---|---|---|---|---|---|
| | CLIP-Score (↑) | Structure Dist (↓) | CLIP-Score (↑) | Structure Dist (↓) | |
| Original Images | 32.50% | - | 16.46% | - | - |
| NTI [28] + P2P | 31.50% | 0.0800 | 12.91% | 0.0191 | 12.8% |
| *DPL* (ours) + P2P | **32.23%** | **0.0312** | **14.29%** | **0.0163** | **87.2%** |

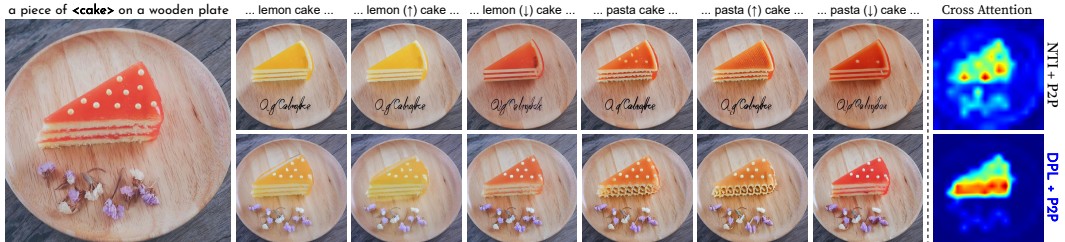

Figure 6: Attention refinement and reweighting. In the given text prompt, only one noun word is learnable. The flowers below the cake are playing as distractors which distort the cross-attention maps while Null-Text Inversion [28] is applied. As a comparison, our method *DPL* successfully filters the cross-attention by our background leakage loss.

Qualitatively, we show word-swap examples on multi-object images in Fig. 7. In contrast to other methods, our proposed approach (*DPL*) effectively addresses the issue of inadvertently editing the background or distractor objects in the provided images.

**Attention refinement and re-weighting.** We also compare NTI and *DPL* by only tuning a single concept token with additional adjective descriptions in the image in Fig. 6. As can be observed, *DPL* keeps better details of the "cake" and modifies its appearance according to the new word. NTI suffers from the cross attention leakage to the background, thus it cannot successfully edit only the "cake" region and distorts the "flowers".

## 5    Conclusions

We presented *DPL* to solve the cross-attention background and distractor object leakage problem in image editing using text-to-image diffusion models. We propose to update the dynamic tokens for each noun word in the prompt in such a way that the resulting cross-attention maps suffer less from attention leakage. The greatly improved cross-attention maps result in considerably better results for text-guided image editing. The experimental results, confirm that *DPL* obtains superior results, especially on complex multi-object scenes. In the Supplementary Material, we further provide a comprehensive discussion on the limitations of our work, its broader impact, and potential directions for future research.

**Limitations and future works.** One limitation of our work is that the smaller cross-attention maps, specifically those with a size of 16×16, contain a greater amount of semantic information compared to maps of larger sizes. While this rich information is beneficial, it limits our ability to achieve precise fine-grained structure control. Furthermore, our current approach has not addressed complex scenarios where a single object in the image corresponds to multiple noun words in a long caption. This remains an unexplored area and presents an opportunity for future research. Additionally, the Stable Diffusion model faces challenges when reconstructing the original image with intricate details due to the compression mechanism in the first stage autoencoder model. The editing of high-frequency information remains a significant and ongoing challenge that requires

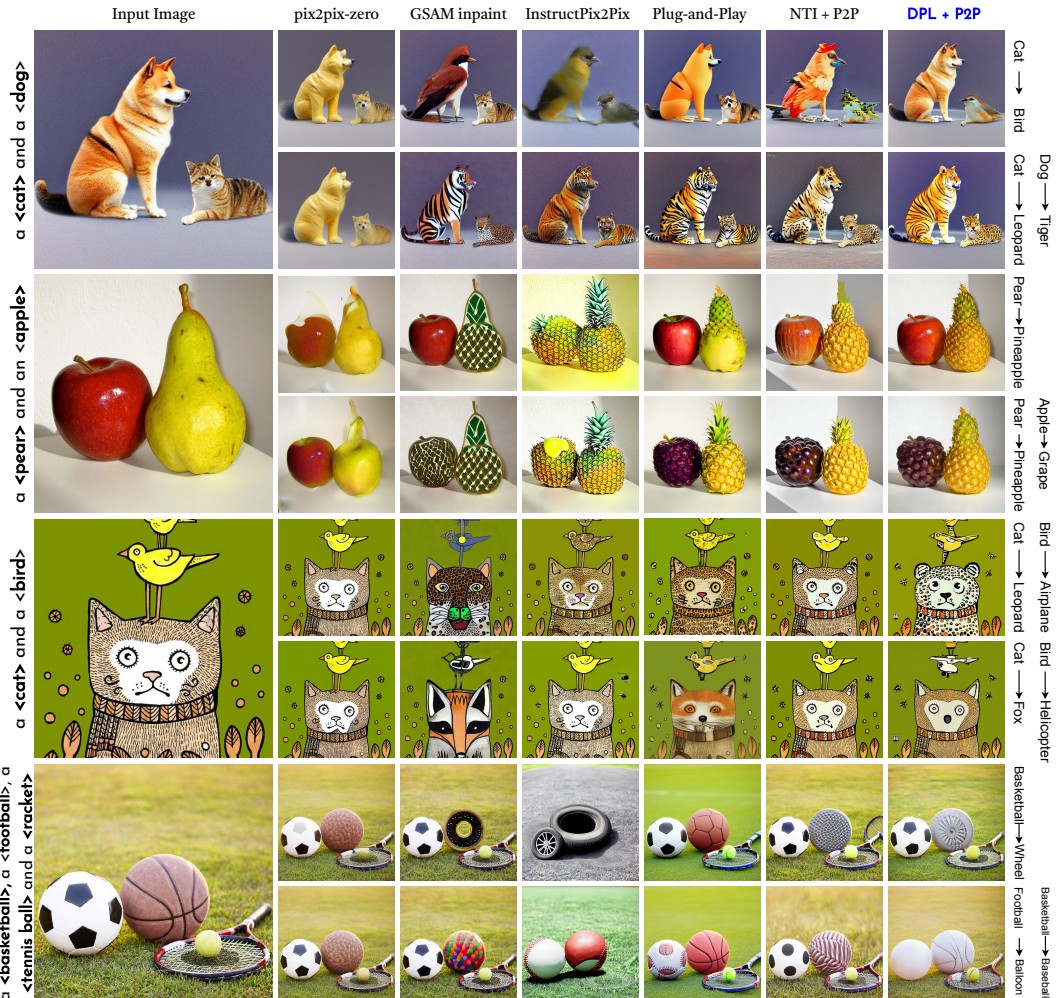

Figure 7: Word-Swap by only modifying the regions corresponding to the target concept. We list the source prompts on the left with *dynamic* tokens in the curly brackets. Each line is corresponding to modifying one or multiple concepts with various methods.

further investigation and development. Addressing these limitations and advancing our understanding in these areas will contribute to the improvement and refinement of image editing techniques.

**Broader impacts.** The application of text-to-image (T2I) models in image editing offers extensive potential for diverse downstream applications, facilitating the adaptation of images to different contexts. The primary objective of our model is to automate and streamline this process, resulting in significant time and resource savings. It is important to acknowledge that current methods have inherent limitations, as discussed in this paper. However, our model can serve as an intermediary solution, expediting the creation process and offering valuable insights for further advancements. It is crucial to remain mindful of potential risks associated with these models, including the dissemination of misinformation, potential for abuse, and introduction of biases. Broader impacts and ethical considerations should be thoroughly addressed and studied in order to responsibly harness the capabilities of such models.

**Acknowledgments.** We acknowledge the support from the Spanish Government funding for projects PID2022-143257NB-I00, TED2021-132513B-I00 funded by MCIN/AEI/10.13039/501100011033 and by FSE+ and the European Union NextGenerationEU/PRTR, and the CERCA Programme of Generalitat de Catalunya. We also thank for the insightful discussion with Juan A Rodriguez, Danna Xue, and Marc Paz.

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
