# Supplementary Material:
# Dynamic Prompt Learning: Addressing Cross-Attention Leakage for Text-Based Image Editing

**Kai Wang**[1], **Fei Yang**[1]*, **Shiqi Yang**[1], **Muhammad Atif Butt**[1], **Joost van de Weijer**[1,2]

[1]Computer Vision Center, Barcelona, Spain

[2] Universitat Autonoma de Barcelona, Barcelona, Spain

{kwang,fyang,syang,mabutt,joost}@cvc.uab.es

## A   Limitations and future works

One limitation of our work is that the smaller cross-attention maps, specifically those with a size of $16 \times 16$, contain a greater amount of semantic information compared to maps of larger sizes. While this rich information is beneficial, it limits our ability to achieve precise fine-grained structure control. Furthermore, our current approach has not addressed complex scenarios where a single object in the image corresponds to multiple noun words in a long caption. This remains an unexplored area and presents an opportunity for future research. Additionally, the Stable Diffusion model faces challenges when reconstructing the original image with intricate details due to the compression mechanism in the first stage autoencoder model. The editing of high-frequency information remains a significant and ongoing challenge that requires further investigation and development. Addressing these limitations and advancing our understanding in these areas will contribute to the improvement and refinement of image editing techniques.

## B   Broader impacts

The application of text-to-image models in image editing offers extensive potential for diverse downstream applications, facilitating the adaptation of images to different contexts. The primary objective of our model is to automate and streamline this process, resulting in significant time and resource savings. It is important to acknowledge that current methods have inherent limitations, as discussed in this paper. However, our model can serve as an intermediary solution, expediting the creation process and offering valuable insights for further advancements. It is crucial to remain mindful of potential risks associated with these models, including the dissemination of misinformation, potential for abuse, and introduction of biases. Broader impacts and ethical considerations should be thoroughly addressed and studied in order to responsibly harness the capabilities of such models.

## C   Dataset protocols

In the context of image retrieval using the CLIP-retrieval tool [1], the objective is to search for images that contain at least two objects, such as a cat and a dog, while ensuring a high CLIP similarity to the given prompt. The input prompts, like "a cat and a dog" guide the search process to retrieve images that meet this criterion, ensuring the presence of both specified objects. The goal is to find images that not only contain the desired objects but also exhibit a strong similarity to the input prompt based on the CLIP model. Through this process, we collected a total of 327 images by conducting 32 different searches with various prompts. Our intention is to release this curated image dataset as a benchmark for future research endeavors. The searching templates include "a {object 1} and a {object 2}", "a

---

*Corresponding author

37th Conference on Neural Information Processing Systems (NeurIPS 2023).

| Prompts | Image Number | Dataset |
| --- | --- | --- |
| a clock and a book | 24 | MO-Set; URS-Set |
| a dog and a bird | 19 | MO-Set; URS-Set |
| a ball and a cat | 19 | MO-Set |
| a book and a pen | 17 | MO-Set |
| a cat and a dog | 16 | MO-Set; URS-Set |
| a knife and a fork | 13 | MO-Set |
| a cat and a bird | 13 | MO-Set; URS-Set |
| a person on a bike | 13 | MO-Set |
| a horse and a sheep | 11 | MO-Set |
| a cake in a plate | 4 | URS-Set |
| a keyboard and a mouse | 4 | MO-Set |
| a cat and a dog on the grass | 3 | MO-Set |
| a piano and a chair in the room | 2 | URS-Set |
| a pear and an apple | 2 | URS-Set |
| a pizza on a table | 2 | URS-Set |

Table 2: The comprehensive dataset statistics from our CLIP-retrieval [1] searching.

{object 1} in/on a {object 2}", "a {object 1} and a {object 2} on a {object 3}", etc. In Table 2, we detailedly present the comprehensive dataset statistics.

# D  Ablation study

**Ablation study for hyperparameters.**  In addition to the ablation study depicted in Fig.5 of the primary manuscript, supplementary ablation experiments were conducted involving the manipulation of hyperparameter combinations. The specific hyperparameter values were determined as follows: $\lambda_{at} = 1.0, \alpha_{at} = 25, \beta_{at} = 0.3$ for attention balancing loss, $\lambda_{dj} = 0.05, \alpha_{dj} = 25, \beta_{dj} = 0.9$ for disjoint object attention loss, and $\lambda_{bg} = 0.05, \alpha_{bg} = 50, \beta_{bg} = 0.7$ for background leakage loss. The outcomes of these additional ablation experiments are visually presented in Fig. 10. Note that these hyperparameters have been selected based on the validation set, and are then applied on test (see Fig.5) and also kept constant for the other images used for illustrations.

Additionally, in order to showcase the effect of each loss component, we visualize the averaged cross-attention maps while progressively adding new loss components in Figure 11. From the visualizations, we can observe that when using only the attention balancing loss ($\mathcal{L}_{at}$), *DPL* still suffers from cross-attention leakage from the background and distractor objects. However, with the inclusion of the disjoint object attention loss ($\mathcal{L}_{dj}$), the leakage from distractor objects is relieved to a certain extent. Finally, the background leakage loss ($\mathcal{L}_{bg}$) effectively filters out the leakage originating from the background, leading to improved attention localization.

**Ablation for the Gradual Optimization technique.**  Regarding "Gradual Optimization for Token Updates," its central aim is to alleviate update pressures at each step of the process, particularly due to the accumulation of cross-attention leakage during denoising. We've introduced a mechanism ensuring all losses attain predefined thresholds at each step, aiming to prevent overfitting of cross-attention maps. Such overfitting could lead to erroneous cross attentions, detrimentally affecting the editing process, as can be observed in Fig. 8. Hence, the gradual optimization strategy enhances robustness and accuracy throughout editing.

**Progressively infusion of the attention maps.**  Imposing strict DPL indeed limits the editing region, which aligns with practical scenarios. Our method *DPL* is to position the generated target object at the source object's original spot while maintaining overall stability. This constraint is akin to attention injection seen in cross-attention [2, 8, 5, 7] and self-attention methods [10, 3]. Adjusting the size of the generated target remains unexplored. Nonetheless, the constraint can be relaxed using partial attention injection as depicted in Fig.6 of P2P [5]. The effect of attention injection across varied steps for DPL is shown in Fig. 9.

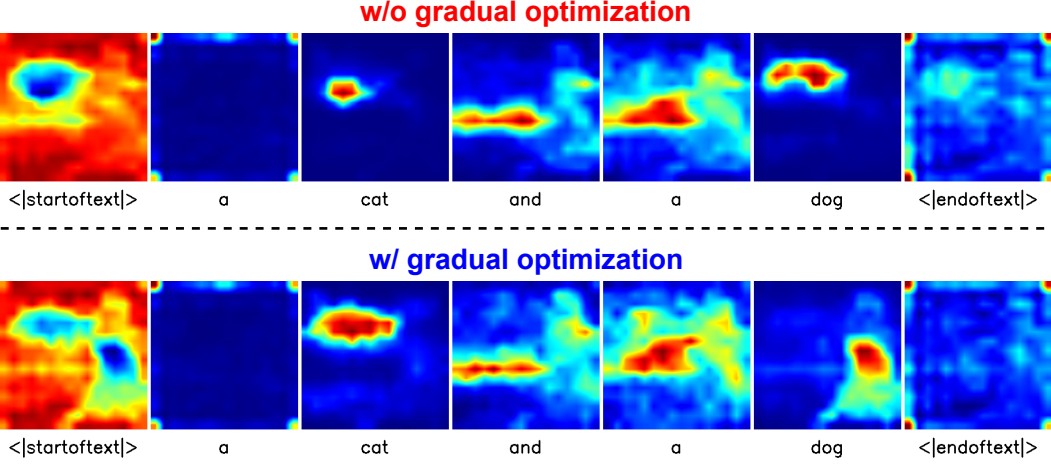

Figure 8: Ablation study of the Gradual Optimization for Token Updates.

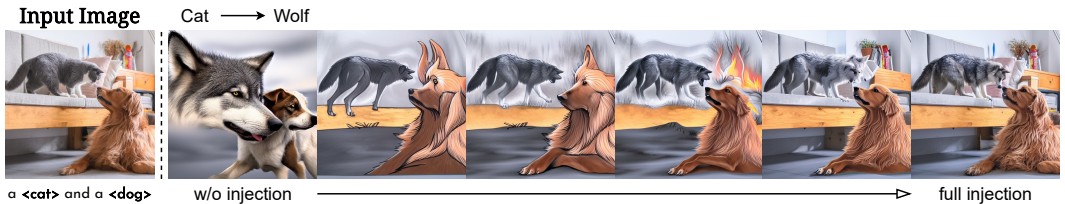

Figure 9: Progressively infusing the attention maps across diverse diffusion steps ranging from 0% (left) to 100% (right) of the steps.

# E   Attention-based background estimation

To provide a comprehensive analysis of the Stable Diffusion UNet module, we extend the DDIM inversion visualization (Fig.3 in the main paper) to include each component within the module. In Fig. 12, we present the visualization of all components for two images: one with a single object and the other with two objects. During the inversion process using the DDIM sampler, we calculate the average representation of each component in the UNet, including self-attention (SA), cross-attention (CA), and feature maps, among others. By visualizing these components, we aim to gain insights into their semantics. Through our observations, we note the following: **(1)** Components with larger sizes tend to pay more attention to high-frequency information. **(2)** Self-attention and feature maps with sizes of $16 \times 16$ and $32 \times 32$ respectively, provide significant insights into the semantic layouts of the image.

Furthermore, we conducted a thorough comparison of all possible combinations using the Intersection over Union (IoU) metric over a collection of single-object images consisting of 40 images with corresponding object masks. We aim to determine the optimal choice for achieving higher foreground intersection (FG-IS) and background intersection (BG-IS) metrics. We found that this was achieved with the combination of self-attention (SA) with a size of 32, cross-attention (CA) with a size

| CA-size | SA | feature | query | key | value |
|---|---|---|---|---|---|
| 16 | 0.768 | 0.768 | 0.755 | 0.794 | 0.580 |
| 32 | 0.780 | 0.774 | 0.734 | 0.767 | 0.746 |
| 64 | 0.802 | 0.767 | 0.748 | 0.742 | 0.729 |

Table 3: Comparing IoU of matching the cross-attention maps to the self-attention or feature over 40 images. The self-attention is superior to the other components.

| | 1 obj. | 2 obj. |
|---|---|---|
| IoU | 0.803 | 0.749 |
| FG-IS | 0.942 | 0.857 |
| BG-IS | 0.859 | 0.923 |

Table 4: Comparison with segmentation metrics for a one-object set (40 images) and a two-object set (20 images).

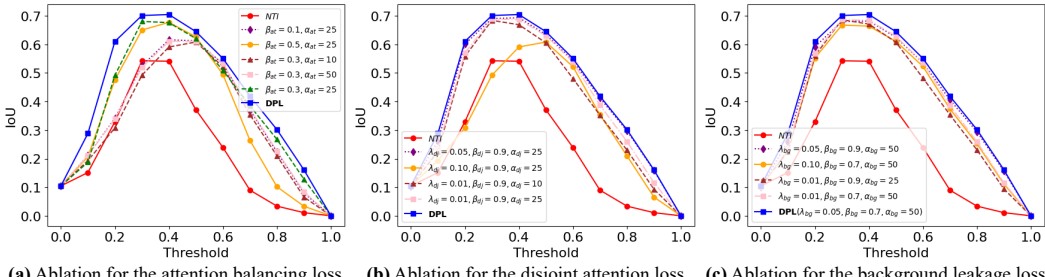

**(a)** Ablation for the attention balancing loss    **(b)** Ablation for the disjoint attention loss    **(c)** Ablation for the background leakage loss

Figure 10: (a) Ablation study over the attention balancing loss with $\lambda_{dj} = 0.0, \lambda_{bg} = 0.0$. (b) Ablation study over the disjoint object attention loss with $(\lambda_{at} = 1.0, \alpha_{at} = 25, \beta_{at} = 0.3), \lambda_{bg} = 0.0$. (c) Ablation over the background leakage loss with $(\lambda_{at} = 1.0, \alpha_{at} = 25, \beta_{at} = 0.3), (\lambda_{dj} = 0.05, \alpha_{dj} = 25, \beta_{dj} = 0.9)$.

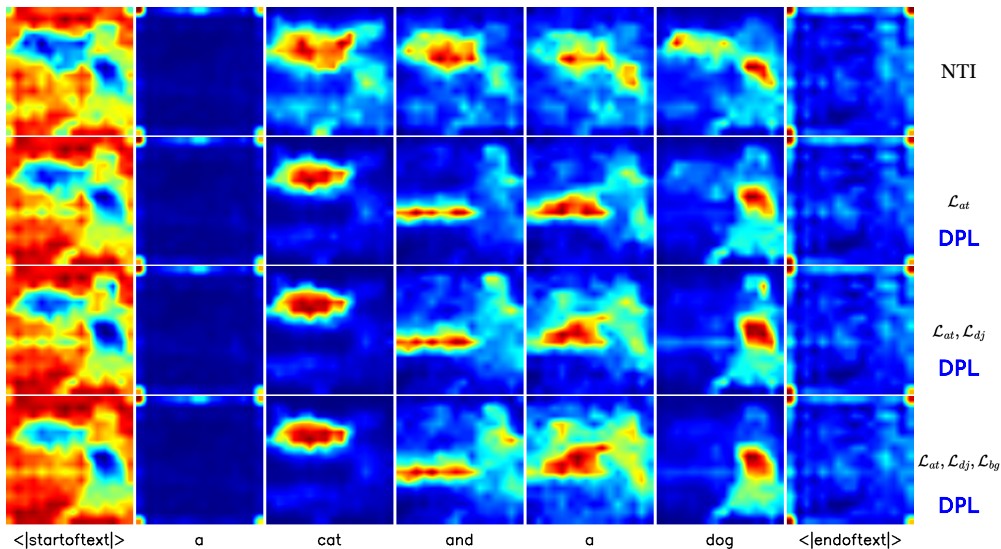

Figure 11: To demonstrate the effectiveness of each loss component, we visualize the average cross-attention maps. By comparing with NTI [7] as the baseline, we can understand the importance of each loss component in guiding the attention mechanism of the model. The input image is the same one which is shown in the lower left in Fig. 12.

of 16, and with $TH = 0.2, V = 5$. Table 3 and Figure 13 provide a partial summary of this comprehensive comparison. Table 3 presents the performance comparison between self-attention and feature maps across various metrics, demonstrating the superiority of self-attention in terms of IoU metric. Figure 13 visually illustrates the effectiveness of the proposed optimal combination in generating more accurate and compact segmentations compared to other combinations. This analysis serves to highlight the importance of selecting the appropriate attention mechanisms and parameters in achieving improved segmentation performance for single-object images.

However, the predicted object masks remain imprecise in multi-object scenarios. This issue can largely be attributed to imperfect cross-attention maps. Nonetheless, our approach continues to perform well with regard to predicting the background, as the CA maps are primarily focused on the foreground (FG) regions. In Table 4, we extract the foreground masks from two image collections, which are with 1 object and 2 objects, respectively. By comparing the IoU, foreground (FG-IS) and background intersection (BG-IS) metrics, it is evident that the IoU and FG-IS values are getting lower when there are two objects in the images, while the BG intersection value remains high. As a consequence, the BG masks are still reliable to relieve the background leakage problem.

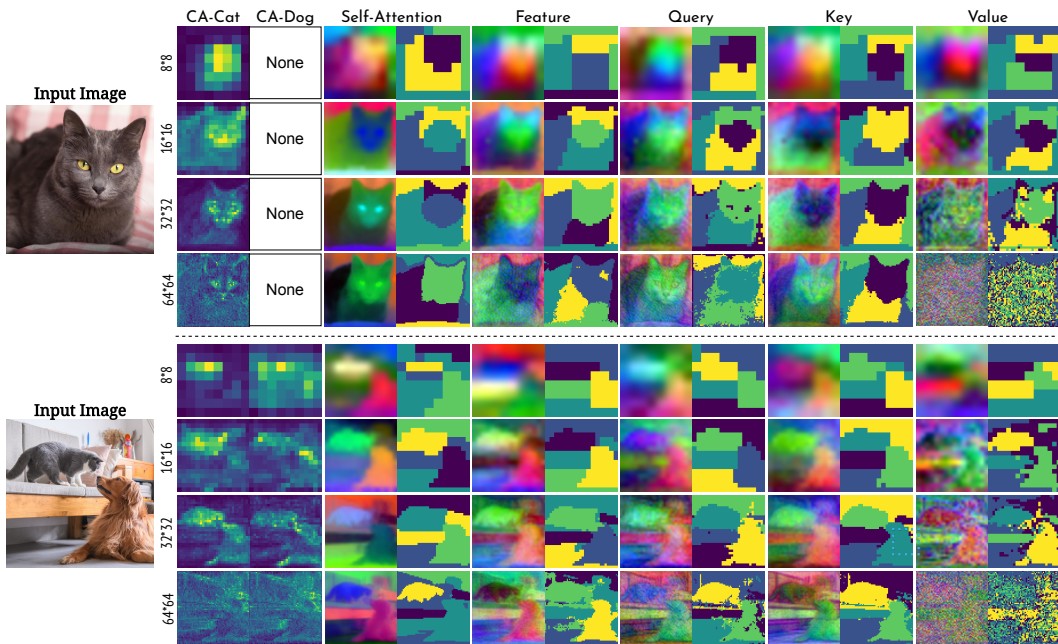

Figure 12: After DDIM inversion, we visualize the average of each component over timestamp $T$ for the input images. From left to right, they are cross-attention (CA) of each noun word, then PCA&Clusters of self-attention (SA), feature, query, key value.

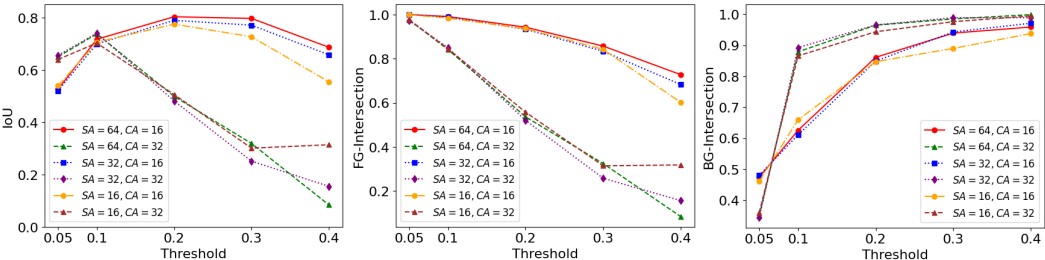

Figure 13: Attention map sizes selection with 40 single-object images for the segmentation matching between self-attention and cross-attention. By comparison, we select $CA = 16, SA = 32, TH = 0.2, V = 5$ as the default hyperparameters of our attention based background estimation.

# F   Cross-attention map comparison with full sentences

In addition to the comparison of *DPL* with NTI [7] solely based on the specified object attention maps in Fig.5, we extend the evaluation by including the comparison with full sentences cross-attention maps, as illustrated in Fig. 14. This comprehensive comparison further demonstrates the adaptability of *DPL* across a wide range of input prompt lengths.

# G   More examples of real image editing

In this section, we provide more samples to extend the comparison with other methods as shown in the main paper. These results are included in Fig. 16 and Fig. 17. Furthermore, extended comparison with DiffEdit [4] and Imagic [6] for image editing is shown in Fig. 15.

## G.1   Word-Swap

In Figure 16, we present additional real image examples showcasing the results of Word-Swap image editing. The objective of this comparison is to highlight the effectiveness of various popular

| method | DPL | NTI [7] | DiffEdit [4] | InstructPix2Pix [2] | pix2pix-zero [8] | Plug-and-Play [10] |
|---|---|---|---|---|---|---|
| User study (%) | 47.1 | 3.8 | 0.9 | 11.1 | 7.3 | 29.8 |

Table 5: Extended user study with additional popular text-guided image editing methods. Based on the evaluation, we can infer that *DPL* and Plug-and-Play are the two approaches that primarily satisfy the evaluators' subjective preferences.

methods when it comes to changing one concept to another in the given images. By focusing on altering a specific concept within each image, we can observe and compare the outcomes produced by these different methods. This analysis provides insights into the performance and capabilities of these methods in accurately accomplishing the desired concept transformation in real image editing scenarios.

### G.2 Attention Refinement and global edit.

In the domain of global image editing, we observe that our method, *DPL*, exhibits less superiority over NTI, as shown in Figure 17. This outcome can be attributed to the fact that *DPL* primarily focuses on object localization within the given image for fine-grained local edits. Consequently, the comparatively limited advantage in overall image details is justified. While our method excels in local edits and object manipulation, it may have some limitations when it comes to global image editing tasks that involve comprehensive modifications across the entire image.

## H Examples of text-guided editing for generated images

In the main paper, our focus was primarily on real image editing tasks to ensure clarity and avoid ambiguity. However, it is worth noting that the issue of cross-attention leakage can also arise in generated images. To demonstrate this, we provide two examples of images generated from the Stable Diffusion model [9] in Figure 18. Similar to the challenges encountered in real image editing examples, we observe that the Null-Text inversion method [7] struggles to accurately locate the corresponding objects and fails to properly edit the desired regions in the generated images. This highlights the significance of addressing the cross-attention leakage problem in both real and generated image editing scenarios, as it can impact the quality and effectiveness of the editing process. While our method, *DPL*, aims to mitigate the cross-attention leakage problem, further research and development are necessary to overcome these challenges in both real and generated image editing tasks.

## I User study

In Figure 19, we provide a screenshot of our user study interface. During the study, participants were presented with two options (NTI and *DPL* anonymously) and asked to evaluate which option better represented the prompt change while maintaining the original image's structure. The goal was to gather user feedback and opinions regarding the effectiveness of the different options in accurately reflecting the intended prompt modifications without altering the underlying structure of the image.

**Extended user study with more comparison methods.** In Table 5, we incorporated with additional popular text-guided image editing methods into the user study, including DiffEdit [4], Instruct-Pix2Pix [2], pix2pix-zero [8] and Plug-and-Play [10]. Based on the evaluation, we can infer that our method, *DPL*, and Plug-and-Play are the two approaches that primarily satisfy the evaluators' subjective preferences. These findings demonstrate the strong performance and user satisfaction of *DPL* and underscore its competitiveness in comparison to the other methods.

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

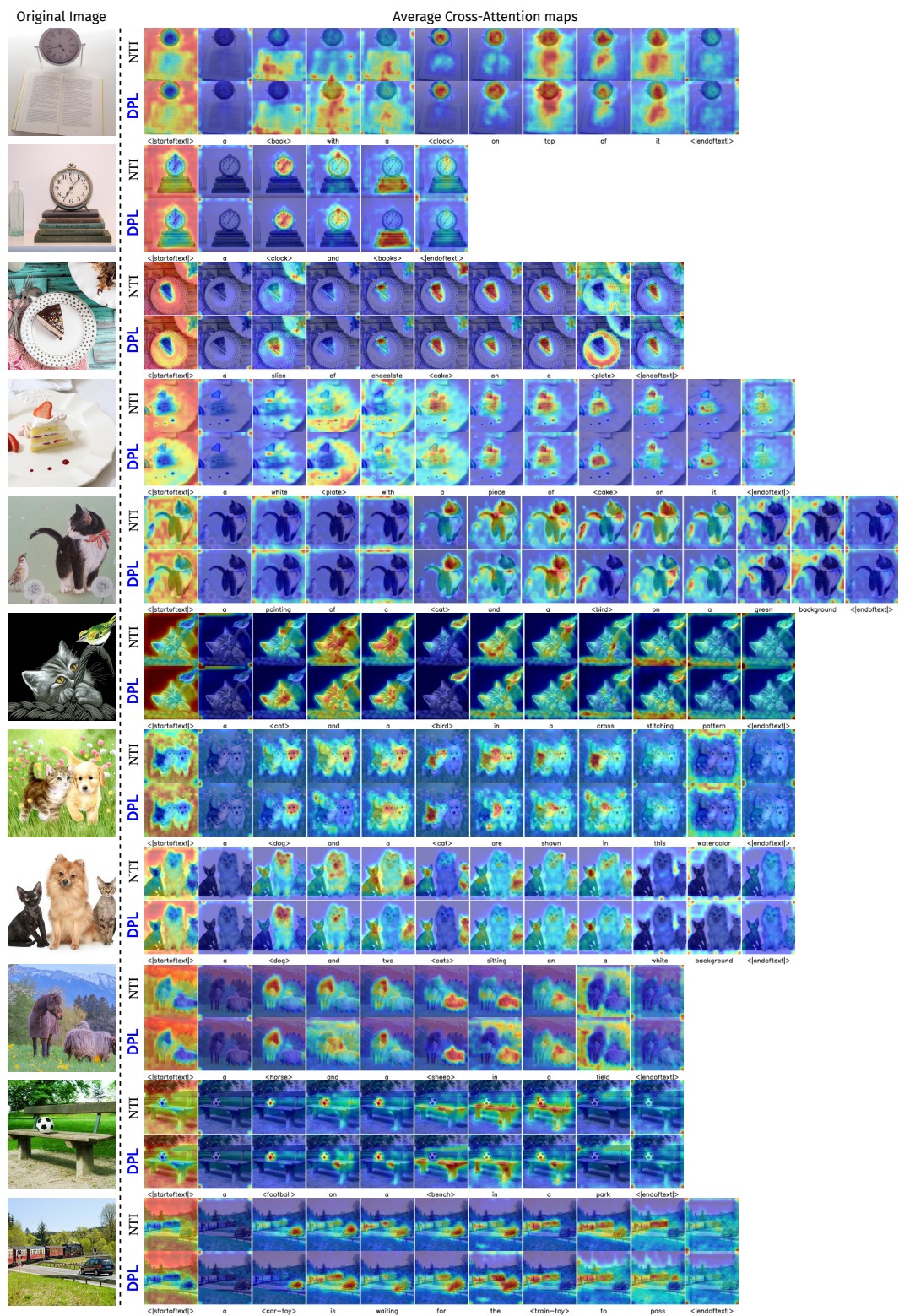

Figure 14: Cross-attention maps comparison with full sentences.

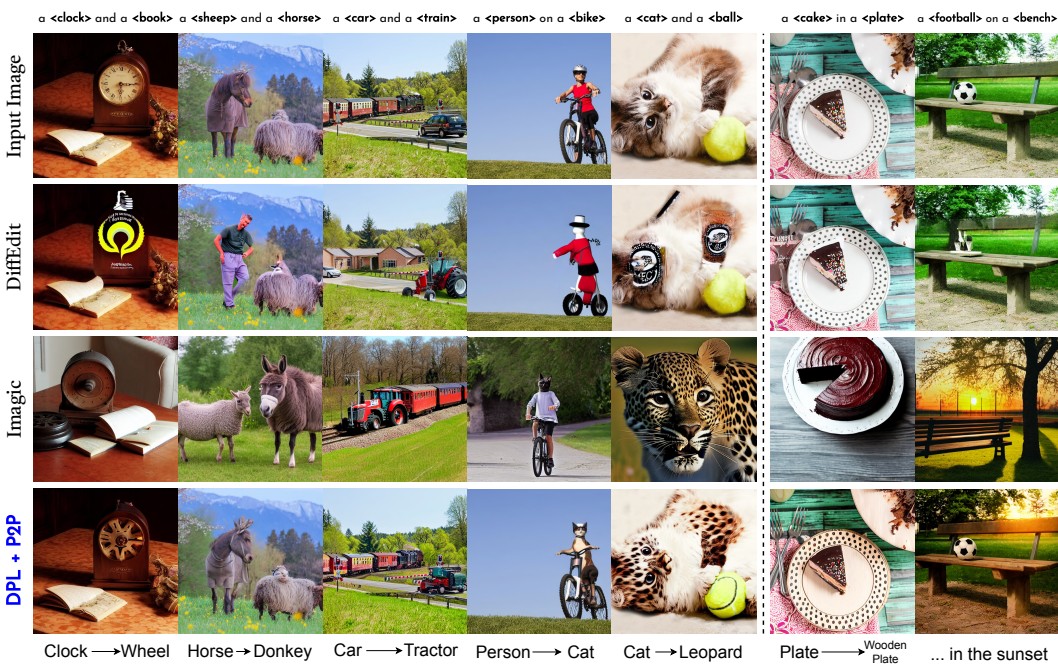

Figure 15: Extended comparison with DiffEdit and Imagic for image editing tasks as Word-Swap (left), Attention Refinement (penultimate column) and Global Editing (last column).

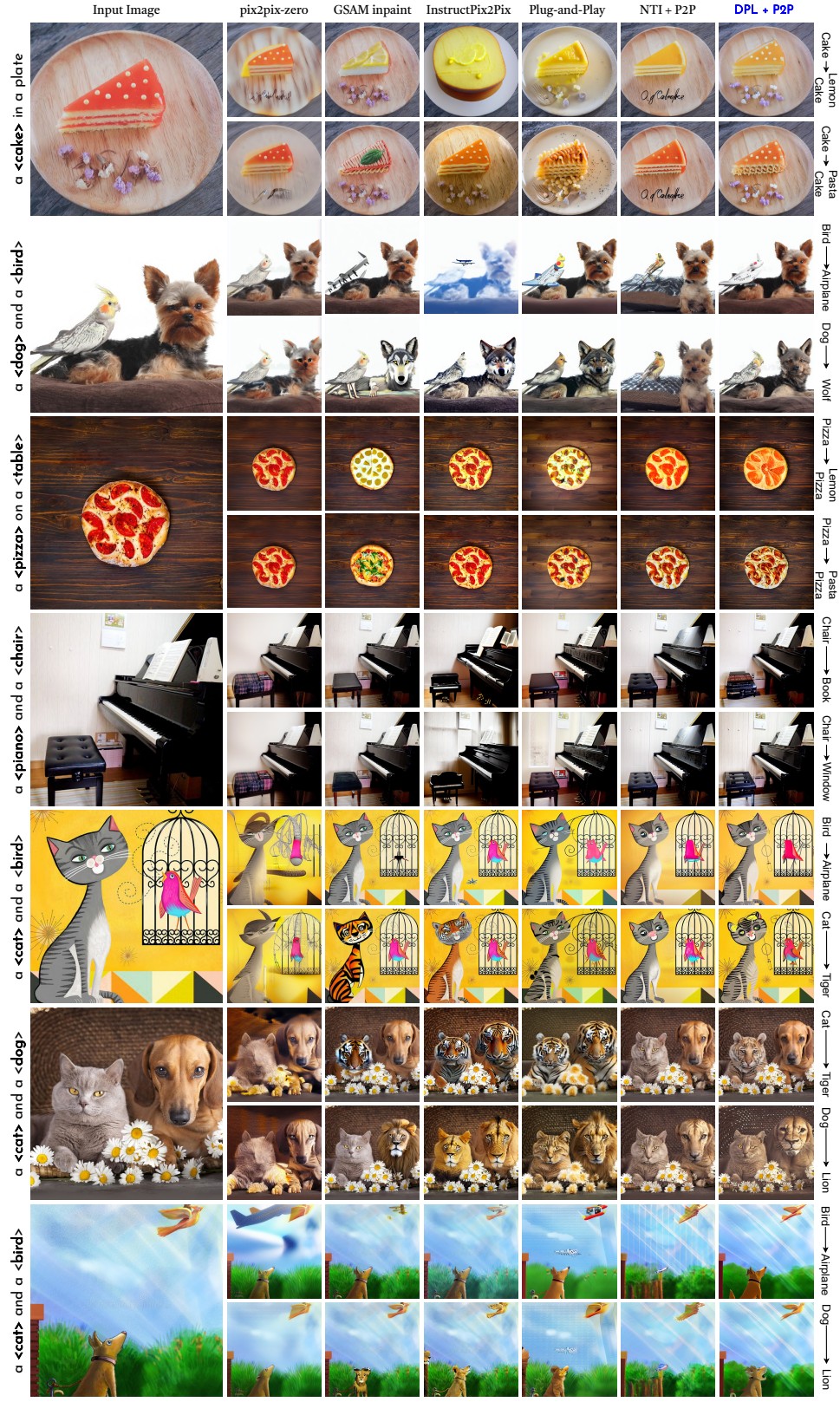

Figure 16: Word-Swap examples with real images.

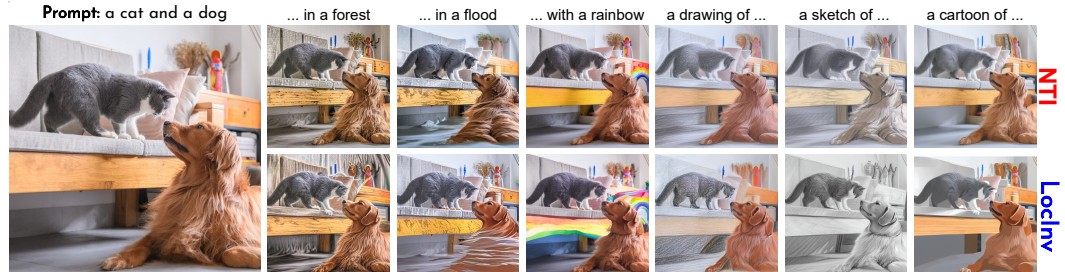

Figure 17: Global editing comparison with NTI [7].

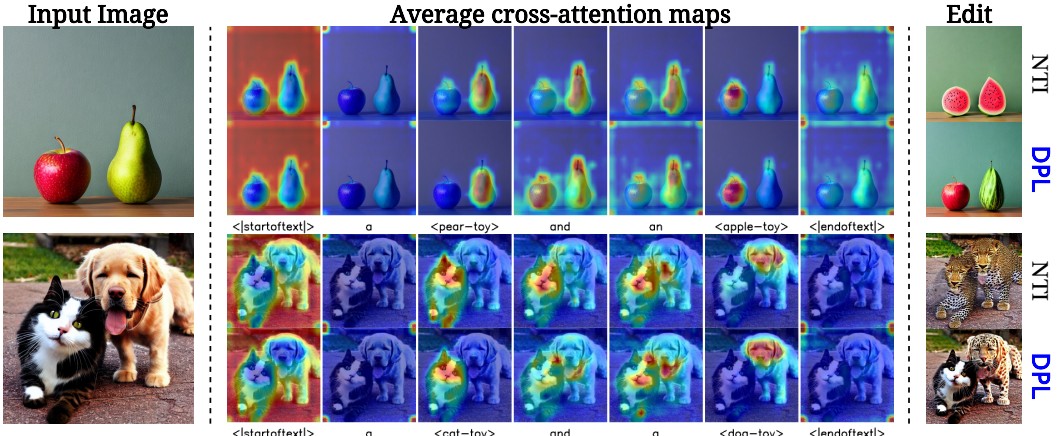

Figure 18: For generated images, the cross-attention leakage problem also presents, even they are generated from Stable Diffusion model [9]. To address this challenge, we conducted experiments involving image editing tasks, specifically transferring *a pear into a watermelon (above)* and *a dog into a leopard (below)*. Notably, our method, *DPL*, demonstrates improved performance in terms of object localization and consistency compared to alternative approaches.

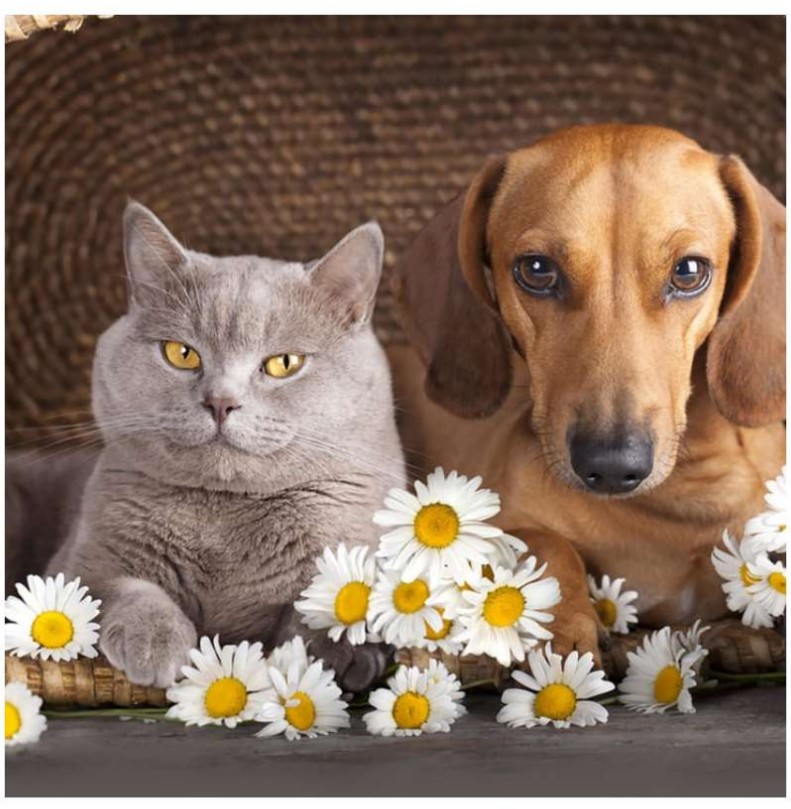

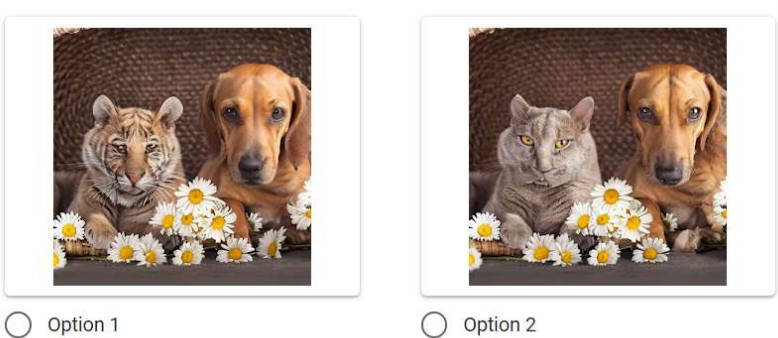

Figure 19: User study print screen. We conduct anonymous user study by mixing the NTI [7] and our method results.