# OpenReview forum: "Dynamic Prompt Learning: Addressing Cross-Attention Leakage for Text-Based Image Editing"
_NeurIPS.cc/2023/Conference — NeurIPS 2023 poster_

### Official Review · Reviewer_zBnt · 2023-07-04

**Soundness:** 3 good
**Presentation:** 3 good
**Contribution:** 2 fair
**Rating:** 5
**Confidence:** 4

**Summary:**

This paper proposes a dynamic prompt learning approach for image editing that modifies self-attentions to more accurately attend to the correct nouns given a text prompt. The proposed approach is used along with null text inversion where the dynamic tokens corresponding with the noun words are updated with a background leakage loss, a disjoint object attention loss and an attention balancing loss. Then p2p is used with the modified cross-attention masks to edit the image. The main benefit of this approach is in editing an image which includes two or more semantically related objects like cat and dog. The authors have collected a set of 60 multi-object images from LAION as a test set and a set of 60 images with semantically related objects for their human evaluations, have shown improvements through qualitative and some quantitative evaluations.

**Strengths:**

1. This paper is well written and explains the method and related works clearly.
2. A novel method as dynamic prompt learning for image editing has been proposed. The paper is mainly focused on a specific axis of image editing where two or more semantically related objects are present in the image and cross attention maps for each object leak to other objects or the background. This approach modifies the attention masks and thus can be used along with p2p for image editing.
3. A test set containing images of multiple objects from LAION-5B is collected for quantitative and qualitative evaluations and a couple of examples are presented in the paper and supplemental showing the superiority of the proposed method.

**Weaknesses:**

1. The scope of the image editing problems this paper aims to improve is narrow mainly useful for cases when there are two or multiple semantically-related objects in the image and for word-swapping editing prompts like cat to dog, cat to tiger, etc.
2. Through only a couple of image editing examples shown in the paper, I am not convinced that this paper is beneficial in general image editing problems where there are not necessarily related objects in the scene or for other types of editing prompts rather than word swapping.
  -  It is unclear what type of editing prompts are included in the collected test set, but from discussions, it looks like it is mainly focusing on word swapping. Also, size of the test set is fairly small, and the studies are all done in the above narrowed down scope in image editing, thus not conclusive.
- The quantitative clip scores are only reported in two scenarios (1) cat → leopard and dog → tiger and (2)  book → box and clock → sunflower and improvements are still marginal.
- On the human evaluation study, I'd expect it to be done on other image editing types with single or multiple objects either semantically related or unrelated, on a larger set, and to compare with other baselines such as instructpix2pix and plug-and-play.
3. On auto evaluations, one potential metric to measure the correctness/accuracy of this method could be applying an object segmentation or object detection method on the edited image, predict label for the edited objects, check if the predicted label matches with the given prompt. Compute the average accuracy score and compare with the baselines.

**Questions:**

1. Please describe your evaluation datasets and their editing prompts more clearly. Is the URS-Set similar to the MO-Set? what is the
difference between these two sets? Since LAION-5B is pretty large, I'd suggest the authors to expand their test set to include more varied image editing prompts and examples.
2. I wonder how much gain this method introduces on image editing problems with one object or two unrelated objects in the image (if any).

**Limitations:**

The authors have neither discussed the limitations of their work, nor its negative societal impact. A broader study on different editing types as mentioned above could clarify the limitations.

---

> ### Author Rebuttal · Authors · 2023-08-09
>
> We appreciate your feedback and will use the discussion to improve. Below we use the references in the main paper.
>
> $\textbf{W1}:$ Actually DPL also works for Word-Swap with less related concepts. As we have shown in Fig.7, Fig.13, Fig.17, we successfully edit the bird into airplane/helicopter, the ball/clock into wheel/balloon, the chair into book/window, a person into a cat, etc. As to the scope of text-guided image editing, P2P[14] is designed to solve prompt-based editing in the denoising phases of a pretrained T2I diffusion models. After, NTI[26] generalized this technique to real-images. Experimenting with these codes, we quickly found that they fail for more complex scene with semantically related objects and background. We consider this one of the main short-comings of T2I and addressing this relevant problem can be important for many realistic image editing applications.
>
> $\textbf{W2.1}:$  It is true that DPL is most beneficial when there are leakage issues in the image. In scenarios where no leakage issues are present, using DPL may not be necessary, as it would incur additional time complexity. DPL can be seen as an augmentation to existing cross-attention based image editing methods[4,14,26]. Its purpose is to improve the accuracy and reliability of cross-attention maps, which is particularly essential for tasks like Word-Swap. For other types of image editing tasks, Fig.6 in the main paper illustrates attention refinement, highlighting the effectiveness of DPL in improving attention maps by filtering the background leakage. Fig.14 in the supplementary material showcases global editing, where DPL may have shown comparatively less superiority. We also include more instances in the rebuttal PDF.
>
> $\textbf{W2.2, Q1}:$  We encountered the challenge of lacking benchmarks tailored for multi-object image editing. We adopted the protocol employed in pix2pix-zero[28] and gathered images from the LAION-5B[37] per our dataset protocols (Supp.C). Through search templates and preprocessing steps, we curated a collection of 327 images from 32 distinct prompts. Finally, we retained only the prompts with more images. This contrasts with NTI and pix2pix-zero datasets, which encompass 100/250 single-object images respectively. Thus, our dataset's scale is considerable. Moreover, our method's efficacy on generated images is evaluated in Supp.H and Fig.15. Here, the number of images can potentially be infinite. This framework permits us to showcase DPL's applicability beyond fixed dataset constraints.
>
> | Prompts | Image Number | Split |
> | -------- | -------- | -------- |
> | a clock and a book | 24 | MO-Set; URS-Set |
> | a dog and a bird | 19 | MO-Set; URS-Set |
> | a ball and a cat | 19 | MO-Set |
> | a book and a pen | 17 | MO-Set |
> | a cat and a dog | 16 | MO-Set; URS-Set |
> | a knife and a fork | 13 | MO-Set |
> | a cat and a bird | 13 | MO-Set; URS-Set |
> | a person on a bike | 13 | MO-Set |
> | a horse and a sheep | 11 | MO-Set |
> | a cake in a plate | 4 | URS-Set |
> | a keyboard and a mouse | 4 | MO-Set |
> | a cat and a dog on the grass | 3 | MO-Set |
> | a piano and a chair in the room | 2 | URS-Set |
> | a pear and an apple | 2 | URS-Set |
> | a pizza on a table | 2 | URS-Set |
>
> $\textbf{W2.3,W2.4}:$ To assess the performance of our editing method, we adopt a similar approach to pix2pix-zero[28], creating two scenarios that demand changes to both objects. However, the improvements achieved in these cases are limited due to the narrow improvement spaces. Additionally, we observe that even the original multi-object images do not closely match the textual prompts, which further impacts the editing outcomes. To gain a more comprehensive evaluation, we conducted a user study that covered a broad spectrum of image editing tasks, going beyond these two setups. As you requested, we extended the user study with more methods as below. Importantly, DPL received higher satisfaction ratings from the participants.
>
> | method | DPL (ours) | NTI | DiffEdit | InstructP2P | Pix2Pix | PnP |
> | :--------: | :--------: |:--------: |:--------: |:--------: |:--------: |:--------: |
> User study (%) | 47.1 | 3.8 | 0.9 | 11.1 | 7.3 | 29.8 |
>
> $\textbf{W3}:$ In our evaluation of editing accuracy, we established ground truths for our dataset images. These ground truths serve to compute IoU with corresponding cross-attention maps from our DPL inversion. This assessment is presented in Fig.5 of the main paper and Fig.8 in the supplementary. Further evaluation encompasses CLIP semantic similarity analysis and a user study, with summarized outcomes in Table 1. As per your request, we incorporated the Segment-Anything model for segmentation map generation based on prompts from edited images. The resultant IoU scores for our DPL method and the baseline NTI stand at 0.48 and 0.37, respectively, assessed over the MO-Set Test-Split. These scores affirm DPL's superior segmentation accuracy compared to the baseline.
>
> $\textbf{Q2}:$ In cases where images feature a single main object, our DPL may exhibit limited improvement over other methods, as depicted in Fig.6 of the main paper and Fig.13 in the supplementary. This observation arises from the fact that in such scenarios, the cross-attention mechanism might lack additional objects to attend to, thus offering minimal influence on the editing process. However, DPL shines in scenarios with multiple disparate objects. Here, DPL excels in effectively filtering both background and distractor object leakage, showcased in Fig.12 and Fig.13 of the supplementary. Furthermore, DPL is specifically tailored to address situations where background and distractor object leakage significantly impact editing performance. It is an augmentation to existing editing methods. We do not assert DPL as a comprehensive solution to all existing editing challenges.
>
> $\textbf{Limitations}:$ We have presented our limitations and broader impacts in Supp.A and Supp.B and aim to move these to the main paper in the future version.

---

> > ### Comment · Reviewer_zBnt · 2023-08-19
> >
> > Thanks to the authors for their response and incorporating my suggestions to have a more solid evaluation. Most of my concerns have been resolved by this rebuttal, and I'd be happy to increase my rating.

---

> > > ### Author Response · Authors · 2023-08-19
> > > **Thanks for the positive response from the reviewer**
> > >
> > > We sincerely appreciate your valuable feedback and raising the rating score. The discussion surely will help us to improve the paper in future versions. Moreover, we are always available for any further discussions to address all of your remaining concerns.

---

### Official Review · Reviewer_m2Wr · 2023-07-05

**Soundness:** 3 good
**Presentation:** 3 good
**Contribution:** 3 good
**Rating:** 6
**Confidence:** 3

**Summary:**

The paper proposes a new method to improve the attention masking for attention-based local editing.

**Strengths:**

The proposed loss functions are novel, intuitive and effectively improves the inversion process of text-to-image diffusion model.

**Weaknesses:**

1. Please show the quantitative evaluation of the object masking performance between the mask generated from Diffedit.

2. Quantitative comparison with baseline models are not enough. Please put more comparison between other baselines such as Plug-and-play and Pix2pix-zero, instructPix2Pix. At least include user study.

3. Does it improve other attention-based editing mechanism?

**Questions:**

Please see the weakness part.

---

> ### Author Rebuttal · Authors · 2023-08-09
>
> We appreciate your feedback and will incorporate the discussions mentioned below to enhance the quality of our paper. Note that we utilize the numerical references to cite sources within the main paper.
>
> $\textbf{W1}:$ As requested, we performed an evaluation of the binary masks obtained from the DiffEdit method on our MO-Set Test-Split. The resulting Intersection over Union (IoU) score for DiffEdit is 0.41, which is similar to NTI but notably lower than our method, DPL (as depicted in Fig. 5-(a)). Moreover, we observed that DiffEdit generated masks with numerous noisy points, as illustrated in Fig. 1. As a consequence, the editing results were not satisfactory according to our user study and qualitative comparisons, as detailed in the author rebuttal PDF. This comparison demonstrates the superior performance of DPL in accurately matching desired editing areas, highlighting its effectiveness in comparison to other methods.
>
> $\textbf{W2}:$ As requested, we incorporated four additional popular text-guided image editing methods into the user study, and the results are presented in the table below. Based on the evaluation, we can infer that our method, DPL, and Plug-and-Play [40] are the two approaches that primarily satisfy the evaluators' subjective preferences. These findings demonstrate the strong performance and user satisfaction of DPL and underscore its competitiveness in comparison to the other methods.
>
> | method | DPL (ours) | Null-Text Inversion [26] | DiffEdit [9] | InstructPix2Pix [4] | Pix2Pix-zero [28] | Plug-and-Play [40] |
> | :--------: | :--------: |:--------: |:--------: |:--------: |:--------: |:--------: |
> User study (%) | 47.1 | 3.8 | 0.9 | 11.1 | 7.3 | 29.8 |
>
> $\textbf{W3}:$ Our method DPL is regularizing the cross attention maps during the denoising phases of the DDIM inversion. As a result, it has the potential to enhance the performance of other existing cross-attention based editing mechanisms.  As of the NeurIPS 2023 deadline, we found only one available method, Null-Text inversion [26], that works on real image inversion and text-guided image editing. Moreover, their image editing approach is based on references from P2P [14].  On the other hand, Pix2pix-zero [28] is also working on cross-attention injection, but it adopts noise-regularization, which can disrupt the reconstruction property of DDIM inversion. Furthermore, Pix2pix-zero is primarily focused on changing the entire image style by discovering edit directions using GPT and CLIP models. In our case, we share more similarities with P2P streams, where we modify textual prompts to guide the editing process.

---

> > ### Comment · Reviewer_m2Wr · 2023-08-17
> >
> > The author rebuttal addressed my concerns. Therefore I keep my score to weak accept.

---

> > > ### Author Response · Authors · 2023-08-17
> > > **Thanks for the positive response from the reviewer**
> > >
> > > We sincerely appreciate all of your efforts to provide your feedback. The discussion will help to improve the paper quality in any future versions.

---

### Official Review · Reviewer_sXMN · 2023-07-06

**Soundness:** 3 good
**Presentation:** 3 good
**Contribution:** 3 good
**Rating:** 7
**Confidence:** 4

**Summary:**

The authors first point out that inferior cross-attention maps regarding noun text tokens are the main causes of failures cases in prompt-based editing methods, such as prompt-to-prompt (P2P).
To tackle this, they propose to optimize noun text features with three objectives at each denoising timestep;
i) minimize the overlap between different cross-attention maps, ii) keep the background mask intact, which is automatically inferred from both cross- and self-attention maps, and iii) keep the max value of cross-attention maps related to nouns high.
This way, they achieve better editing results with cross-attention maps that are better aligned with the object regions.

**Strengths:**

- the authors tackle a meaningful task
- it is easy to understand the motivation and method with various visualizations
- the proposed method presents improved performance on a variety of evaluation metrics

**Weaknesses:**

- increased inference time
- only can solve cases related to noun text tokens representing the type of object

**Questions:**

- regarding the algorithm 1, isn’t it better to use updated text features by implementing the line 9 after the line 13?
- the thresholds defined by Equation 10 appear to prevent overfitting of the model. what happens if you don't use these thresholds?

**Limitations:**

please refer to the Weaknesses section

---

> ### Author Rebuttal · Authors · 2023-08-09
>
> We appreciate your feedback and will incorporate the discussions mentioned below to enhance the quality of our paper. Note that we utilize the numerical references to cite sources within the main paper and any references not included in the main paper are provided in a list at the end of this response in alphabetical order.
>
> $\textbf{W1}:$ $\textbf{(1)}$ In the training phase, the DPL method focuses solely on updating the token embeddings that are registered in the language model dictionary, without updating the UNet of the Stable Diffusion models [33]. More specifically, DPL exhibits an extended training duration relative to Null-Text Inversion [26] due to our objective of updating token embeddings to enforce regularization on attention maps for specific objects. In our single-GPU experiments using the A40, NTI requires 130 seconds to process one image, while DPL necessitates approximately 280 seconds. $\textbf{(2)}$ However, during inference, both DPL and NTI do not incur any discernible additional time overhead compared to the fundamental Stable Diffusion model [33].
>
> $\textbf{W2}:$ The reviewer is right to point this out (and we will improve our limitations section):
> Indeed, there have already been papers related to image editing that involve verbs. They typically necessitate fine-tuning the T2I models [5,18], which aligns with our approach and runs parallel to our contributions. As a common practice, we freeze the T2I models, similar to previous works [4,9,14,26,28,40], to prevent the models from overfitting to a specific image. However, it is essential to acknowledge that freezing the T2I models comes with some shortcomings, as pointed out by the reviewer. These limitations should be carefully considered when evaluating the overall performance and effectiveness. There are also certain papers [A,B] focusing on removing objects or inpainting using user-defined masks [1,27,42] in given images. However, the exploration of adding an object in an appropriate position without relying on a mask is currently a very active research area. This direction typically requires model fine-tuning to achieve desirable results and is a parallel research to text-guided image editing as our method DPL. We will include a discussion on this in our limitations section.
>
> $\textbf{Q1}:$ Thanks for pointing it out. You are correct here. The text conditions' initialization appears on line 9. Then the updating of token representations causes changes in the text conditions. To avoid misunderstandings, it is preferable to add another line to describe this process after Line 13 in the future version.
>
> $\textbf{Q2}:$ By using Equation 10, we aim to prevent overfitting of the cross-attention maps, which could otherwise lead to erroneous cross attentions and adversely impact the editing process, as can be observed in Fig.18 of our rebuttal PDF. The gradual optimization approach is thus strategically designed to enhance the robustness and accuracy of our method throughout the editing procedure.
>
> [A] Ablating Concepts in Text-to-Image Diffusion Models. ICCV 2023
>
> [B] Erasing Concepts from Diffusion Models. ICCV 2023

---

### Official Review · Reviewer_qcZn · 2023-07-07

**Soundness:** 3 good
**Presentation:** 3 good
**Contribution:** 3 good
**Rating:** 5
**Confidence:** 4

**Summary:**

The paper propose DPL to solve the cross-attention background and distractor object leakage problem in
image editing using text-to-image diffusion models. The presentation is well written and easy to follow. The discussion and analysis is extensive and interesting. But the experiment dataset is too small to clarify the robustness of this method.

**Strengths:**

1. The presentation is well written and easy to follow.
2. The discussion and analysis is extensive and interesting.
3. The framwork to solve the cross-attention background and distractor object leakage problem is novel.

**Weaknesses:**

1. The experiment dataset is too small to clarify the robustness of this method.

**Questions:**

null

**Limitations:**

null

---

> ### Author Rebuttal · Authors · 2023-08-09
>
> We appreciate your feedback and will incorporate the discussions mentioned below to enhance the quality of our paper. Note that we utilize the numerical references to cite sources within the main paper.
>
> $\textbf{W1}:$ For the creation of our multi-object real-image dataset, we faced the challenge of the absence of standardized benchmarks specifically tailored to real-image text-guided editing tasks. To overcome this limitation, we adopted the protocol used in the pix2pix-zero [28] framework. In doing so, we retrieved images from the LAION-5B dataset [37] as detailed in our dataset protocols section (Supp.C in the supplementary material). With various predefined search templates followed by necessary preprocessing (including watermark removal and manual selection of images without complete objects, etc.), we collected a total of 327 images from 32 different searches. However, these 32 prompts have image numbers ranging from 1 to 37. Therefore, for a meaningful comparison with other methods, we selected only the prompts with more images for our MO-Set and URS-Set, and the others are serving as qualitative candidates, which we already included in Fig.5, Fig.7, Fig.12, and Fig.13. The detailed information is listed in the table below.
> In comparison, the NTI [26], pix2pix-zero [28] datasets contain 100/250 single-object images respectively. Therefore, our data scale is not small in text-guided image editing problems, and we aim to publish our created dataset as a new benchmark for public usage in the future.
> Furthermore, in our evaluations, we also assess the effectiveness of our proposed method, DPL, on generated images. This is demonstrated in Supp.H and Fig.15. In this context, the number of images can potentially be infinite due to the generative nature of the task. This allows us to test and showcase the versatility and applicability of our method beyond the constraints of a fixed dataset.
> The comprehensive dataset statistics are presented as follows:
>
> | Prompts | Image Number | Split |
> | -------- | -------- | -------- |
> | a clock and a book | 24 | MO-Set; URS-Set |
> | a dog and a bird | 19 | MO-Set; URS-Set |
> | a ball and a cat | 19 | MO-Set |
> | a book and a pen | 17 | MO-Set |
> | a cat and a dog | 16 | MO-Set; URS-Set |
> | a knife and a fork | 13 | MO-Set |
> | a cat and a bird | 13 | MO-Set; URS-Set |
> | a person on a bike | 13 | MO-Set |
> | a horse and a sheep | 11 | MO-Set |
> | a cake in a plate | 4 | URS-Set |
> | a keyboard and a mouse | 4 | MO-Set |
> | a cat and a dog on the grass | 3 | MO-Set |
> | a piano and a chair in the room | 2 | URS-Set |
> | a pear and an apple | 2 | URS-Set |
> | a pizza on a table | 2 | URS-Set |

---

> > ### Comment · Reviewer_qcZn · 2023-08-16
> >
> > Thanks for the rebuttal. Some of my concerns have been addressed. I'll keep my original rating.

---

> > > ### Author Response · Authors · 2023-08-16
> > > **Thanks for the positive response from the reviewer**
> > >
> > > We sincerely appreciate your time and efforts to provide your insightful feedback. And the discussion will help us to improve the paper in any future versions.

---

### Official Review · Reviewer_XeHW · 2023-07-10

**Soundness:** 3 good
**Presentation:** 2 fair
**Contribution:** 2 fair
**Rating:** 4
**Confidence:** 5

**Summary:**

This paper proposes Dynamic Prompt Learning (DPL) to address the cross-attention leakage issue for text-based image editing. Based on the observation that inaccurate cross-attention maps cause unintended modifications of regions outside of the targeted area for text-based image editing, the authors propose Dynamic Prompt Learning to force cross-attention maps to focus on correct noun words in the text prompt. By updating the dynamic tokens for nouns in the textual input with the proposed leakage repairment losses, the proposed approach achieves fine-grained image editing over particular objects while preventing undesired changes to other image regions. Experiments of Stable Diffusion models are conducted for word-swap, prompt refinement, and attention re-weighting.

**Strengths:**

1. This paper proposes an interesting approach to address the attention leakage problem for text-based image editing. By adding constraints on the cross-attention maps, the proposed approach regularizes the attention regions of the diffusion models and thus achieves detail-preserving and high-fidelity image editing.
2. The proposed approach is well-motivated and the discovery of the attention leakage problem is interesting and inspiring.
3. The proposed approach achieves better text-based image editing performance on various editing scenarios, as demonstrated in the extensive experiments.

**Weaknesses:**

1. The proposed DPL approach seems like a combination of several components (including the Disjoint Object Attention Loss, Background Leakage Loss, Attention Balancing Loss, Gradual Optimization for Token Updates, and the trick from null-text inversion), and it is not clear about the contribution and necessity of each component. More ablation study is needed to demonstrate the necessity of combining those methods and loss functions.
2. It is not shown how accurate and how robust the attention-based background estimation is. When does the background estimation fail and how this will affect the model if this part fails? The choice of hyperparameters (self-attention with size 32, cross-attention with size 16 and TH = 0.2, V = 5) for this subsection is based on empirical study, and ablation study is not presented.
3. The writing can be improved. The structure of section 3 can be organized more logically and more clearly. The paragraph "Gradual Optimization for Token Updates" is unclear to me.
4. Experiments are conducted on a self-built dataset with a small size (only 100 images for ablation study and 60 images for user study).

**Questions:**

Please refer to the weakness section.

**Limitations:**

The authors addressed the limitations adequately.

---

> ### Author Rebuttal · Authors · 2023-08-09
>
> We appreciate your feedback and will incorporate the discussions mentioned below to enhance the quality of our paper. Note that we utilize the numerical references to cite sources within the main paper.
>
> $\textbf{W1}:$ This study primarily addresses cross-attention leakage issues in the realm of real image editing, encompassing both background and distractor object leakage concerns. These issues have been insufficiently tackled by existing methods. To effectively mitigate leakage, we propose dynamic prompt learning (DPL), aiming to alleviate cross-attention leakage's influence on the editing process. To combat distractor object leakage, we introduce specialized loss functions: Disjoint Object Attention loss and Attention Balancing loss. These are meticulously designed to improve object focus and attention distribution balance. Background leakage is addressed through the Background Leakage Loss. Ablation of each loss component is detailed in Section 4.1 (Fig. 5-(a)) of the main paper, revealing that standalone use of the disjoint object attention loss and background leakage loss underperforms compared to NTI [26]. Combining them is key for enhanced performance. Moreover, the attention balancing loss enhances cross-attention quality, especially when coupled with the other two proposed losses. While detailed hyperparameter ablation is presented in Supp.D, Fig.8, and Fig.9 of the supplementary material, providing a comprehensive analysis of various experimental setups and corresponding outcomes.
>
> $\textbf{W2}:$ Due to page limits we decided to move this part to the supplementary section (Supp.E); we regret any resulting misunderstandings. To address the critical aspect of background estimation in our method, we have extensively investigated this component in Supp.E of the supplementary material. The success of DPL in filtering background leakage relies heavily on accurate background estimation. Hence, we have thoroughly examined various factors, including attention size, feature components, threshold values, and other relevant parameters, to ensure the robustness and reliability of this essential component of our approach.
>
> $\textbf{W3}:$ Given the multifaceted nature of diffusion-based text-guided image editing, we deeply regret any potential confusion arising from our content organization. Regarding "Gradual Optimization for Token Updates," its central aim is to alleviate update pressures at each step of the process, particularly due to the accumulation of cross-attention leakage during denoising. We've introduced a mechanism ensuring all losses attain predefined thresholds at each step, aiming to prevent overfitting of cross-attention maps. Such overfitting could lead to erroneous cross attentions, detrimentally affecting the editing process, as can be observed in Fig.18 of our rebuttal PDF. Hence, the gradual optimization strategy enhances robustness and accuracy throughout editing.
>
> $\textbf{W4}:$ When constructing our multi-object real-image dataset, we encountered the challenge of lacking standardized benchmarks tailored to real-image text-guided editing. To overcome this limitation, we adopted the protocol employed in the pix2pix-zero framework [28]. We gathered images from the LAION-5B dataset [37] per our dataset protocols (Supp.C in the supplementary material). Through predefined search templates, along with requisite preprocessing steps (like watermark removal and manual image selection without complete objects), we curated a collection of 327 images derived from 32 distinct prompts. However, these 32 prompts involve image numbers spanning from 1 to 37. Thus, for meaningful comparison with other methods, we retained only the prompts with higher image counts for our MO-Set and URS-Set. The remaining ones serve as qualitative candidates, showcased in Fig.5, Fig.7, Fig.12, Fig.13, etc. This contrasts with NTI[26] and pix2pix-zero[28] datasets, which encompass 100/250 single-object images respectively. Our dataset's scale is thus considerable within the context of text-guided image editing problems. Furthermore, our method's efficacy on generated images is evaluated, as demonstrated in Supp.H and Fig.15. Here, the number of images can potentially be infinite due to the generative nature of the T2I models. This framework permits us to test and showcase our method's versatility and applicability beyond fixed dataset constraints.
> The comprehensive dataset statistics are presented as follows:
>
> | Prompts | Image Number | Dataset |
> | -------- | -------- | -------- |
> | a clock and a book | 24 | MO-Set; URS-Set |
> | a dog and a bird | 19 | MO-Set; URS-Set |
> | a ball and a cat | 19 | MO-Set |
> | a book and a pen | 17 | MO-Set |
> | a cat and a dog | 16 | MO-Set; URS-Set |
> | a knife and a fork | 13 | MO-Set |
> | a cat and a bird | 13 | MO-Set; URS-Set |
> | a person on a bike | 13 | MO-Set |
> | a horse and a sheep | 11 | MO-Set |
> | a cake in a plate | 4 | URS-Set |
> | a keyboard and a mouse | 4 | MO-Set |
> | a cat and a dog on the grass | 3 | MO-Set |
> | a piano and a chair in the room | 2 | URS-Set |
> | a pear and an apple | 2 | URS-Set |
> | a pizza on a table | 2 | URS-Set |

---

> ### Comment · Reviewer_XeHW · 2023-08-18
> **Official comment by reviewer XeHW**
>
> Thank the authors for the rebuttal. The authors addressed some of my questions so I have increased my rating.

---

> > ### Author Response · Authors · 2023-08-18
> > **Thanks for the positive response from the reviewer and we are available for any further discussions**
> >
> > We sincerely appreciate your valuable feedback and raising the evaluation score. We believe that the discussion will help us to improve the paper quality in any future versions. Furthermore, we will always be available for any further discussions if you have any remaining concerns or further inquiries. Your continued engagement is greatly appreciated and contributes to the ongoing improvement of our research.

---

### Official Review · Reviewer_4SLV · 2023-07-19

**Soundness:** 4 excellent
**Presentation:** 3 good
**Contribution:** 3 good
**Rating:** 5
**Confidence:** 5

**Summary:**

This paper works on the fidelity problem (i.e., 'unintended changes of background and distractor objects') in text-based image editing of text-to-image diffusion models. The authors attribute the fidelity problem to cross-attention leakage and propose dynamic prompt learning to force cross-attention maps to focus on correct noun words in the text prompt. The paper shows a comparison with a couple of approaches, comprehensive image editing evaluations and ablations.

**Strengths:**

1. The fidelity problem that this work focuses on is important. The overall idea of the paper is really simple and seems effective to this problem.

2. The authors present a good number of experiments validating the effectiveness of their approach.  The results from the paper are promising. The editing quality is decent with observable changes.

3. The paper is overall well-written and easy to follow.

**Weaknesses:**


1. This work has the following potential limitations:
- The attention map limits the size of the generated target, as shown in Figure 7.
- Not applicable to the scenarios in that the noun describes the background, or changes are reflected in the verb word, such as adding an item or deleting objectives. Thus, more experiments in the above scenarios should be provided and discussed.
- Not training-free and may have high latency in practical applications.
2. Both qualitative and quantitative results miss comparison with important methods such as DiffEdit [1], Imagic [2], StructureDiffusion [3].

3. The manuscript lacks explanations for some symbols, such as symbols in Equation (6). It would be great if the author could highlight these symbols in the diagram to refine the explanation of the method.

[1] Diffedit: Diffusion based semantic image editing with mask guidance. ICLR 2023.

[2] Imagic: Text-based real image editing with diffusion models. CVPR 2023.

[3] Training-Free Structured Diffusion Guidance for Compositional Text-to-Image Synthesis. ICLR 2023.

**Questions:**

1. Please clarify the similarities and differences between DPL and Imagic [2]. In my opinion, both are looking for optimized text embedding, while Imagic seems to remain open to control objectives, while DPL adds too many restrictions through attention maps.

2. How long do training and inference take in the proposed method? Can the authors detail the advantages of DPL compared to the training-free methods (e.g., StructureDiffusion [3])?

3. What are the dimensions of the attention maps, the query/key matrix, and the deep features of the noisy image.  It will be great if the authors could indicate those dimensions in the paper.

**Limitations:**

The authors have provided discussion on the limitations and broader impact in the Supplementary Material. However, as mentioned in the weaknesses, this work has the following potential limitations:
- 1) The attention map limits the size of the generated target.
- 2) Not applicable to the scenarios that the noun describes the background, or changes are reflected in verb word, such as adding an item or deleting objectives.
- 3) Not training-free and may have high latency in practical applications.

---

> ### Author Rebuttal · Authors · 2023-08-09
>
> We appreciate your feedback and will integrate the discussions to improve. We use citations for references in the main paper and any omitted references are listed below.
>
> $\textbf{W1.1}:$ Imposing strict DPL indeed limits the editing region, which aligns with practical scenarios. Our goal is to position the generated target object at the source object's original spot while maintaining overall stability. This constraint is akin to attention injection seen in cross-attention[4,9,14,26,28] and self-attention methods[5,40]. Adjusting the size of the generated target remains unexplored. Nonetheless, the constraint can be relaxed using partial attention injection as depicted in Fig.6 of P2P[14]. The effect of attention injection across varied steps for DPL is shown in Fig. 19 of the rebuttal PDF.
>
> $\textbf{W1.2}:$ Your observations are correct: $\textbf{(1)}$ In the context of background descriptions, the current research refers to it as “global editing”. While we've included global editing performance in Fig.14 of the supplementary, our main emphasis is on identifying object positions corresponding to noun words. Consequently, we don't exhibit significant advancements in this case. $\textbf{(2)}$ Editing involving verb words often requires T2I model fine-tuning, as seen in [5,18], aligning with our approach as parallel contributions. Like previous works [4,9,14,26,28,40], we freeze T2I models to prevent overfitting to specific images. However, it's important to recognize the limitations associated with model freezing. $\textbf{(3)}$ Some papers [A,B] have concentrated on object removal or inpainting using user-defined masks[1,27,42]. In contrast, the act of adding an object in a suitable position without relying on a mask is an active domain. This direction usually entails fine-tuning models for optimal outcomes and runs parallel to DPL.
>
> $\textbf{W1.3}:$ Training-free methods[D,E] focus on augmenting T2I models to enable compositional generation but lack capabilities for image editing. In contrast, text-guided editing methods (NTI[26], DPL, etc.) drastically reduce editing time from hours of human editing to mere minutes. Consequently, these techniques find suitability in practical domains such as creative arts and publicity editing. However, we agree that these techniques are not applicable for real-time editing.
>
> $\textbf{W2}:$ DiffEdit[9] proves unsatisfactory for the cases under our consideration, as evident in Fig. 1. Thus, a direct comparison is omitted from the main paper. Nonetheless, we append a comprehensive DiffEdit comparison in the rebuttal PDF, along with an extended user study below. Note that DPL is the most preferred method (by 47.1%). Regarding StructureDiffusion[D], it is essential to clarify that this is a generation method and not designed for image editing. Imagic[18] is a closed-source approach focusing on non-rigid object variation generation, necessitating per-image T2I model fine-tuning. Our method aligns with text-guided editing pipelines[9,14,26,28,40] by freezing T2I models. Notably, Imagic takes about 20 minutes per image on an A40 GPU, while our DPL achieves approximately 4.5 minutes. Qualitative Imagic comparisons are included in our rebuttal PDF. These clarifications aim to underscore the differences between our method and the aforementioned approaches.
>
> | method | DPL (ours) | NTI | DiffEdit | InstructP2P | Pix2Pix | PnP |
> | :--------: | :--------: |:--------: |:--------: |:--------: |:--------: |:--------: |
> User study (%) | 47.1 | 3.8 | 0.9 | 11.1 | 7.3 | 29.8 |
>
> $\textbf{Q1}:$ It is correct that DPL and Imagic share similarities in updating text embeddings, but it is crucial to emphasize the differences: $\textbf{(1)}$ DPL updates specific tokens in the CLIP dictionary, which grants it better fine-grained control over individual objects. On the other hand, Imagic updates the entire text embedding, which may result in limited control over individual objects, especially in complex scenes with multiple objects. $\textbf{(2)}$ DPL indeed updates only the token embeddings in the CLIP dictionary. In contrast, Imagic requires updating the entire T2I model per image, which is computationally expensive and potentially catastrophic forgetting when the T2I models are finetuned[20,35]. $\textbf{(3)}$ DPL is designed to address a more general text-guided editing task. By comparison, Imagic focuses on non-rigid editing while keeping the same object, e.g., it can not be applied for Word-Swap.
>
> $\textbf{Q2}:$ In training, our DPL exclusively focuses on updating token embeddings, leaving the UNet of the LDM untouched. Notably, DPL's extended training duration, relative to NTI[26], stems from the intent to regulate attention maps. In our experiments, NTI takes 2 minutes per image, while DPL requires about 4.5 minutes. However, both methods introduce no noticeable inference time overhead compared to the base LDM. Diverging from training-free methods[C,D] augmented on LDM, which primarily target compositional generation, DPL stands distinct. Aligning with previous pipelines [9,18,26,28,40], we empower tangible image editing driven by textual cues. Notably, DPL's ability for specific region editing further distinguishes it from existing text-guided editing methods.
>
> $\textbf{Q3}:$ The attention map employed in DPL is 16x16, as indicated in Fig.1, Line 186, and Supp.A. The determination of this cross attention size adheres to practices from prior works[4,14,26]. Moreover, our investigation into various attention sizes is shown in Fig. 3 and the supplementary. Notably, this 16x16 cross attention encompasses QKV and feature shapes of 16x16x160 and 16x16x1280, respectively.
>
> [A] Ablating Concepts in Text-to-Image Diffusion Models. ICCV 2023
>
> [B] Erasing Concepts from Diffusion Models. ICCV 2023
>
> [C] Compositional visual generation with composable diffusion models. ECCV 2022
>
> [D] Training-Free Structured Diffusion Guidance for Compositional Text-to-Image Synthesis. ICLR 2023.

---

> > ### Comment · Reviewer_4SLV · 2023-08-16
> >
> > The authors acknowledge the limitations of their work and also explain their strengths. The authors have addressed most of my concerns and I am willing to keep my original rating.

---

> > > ### Author Response · Authors · 2023-08-16
> > > **Thanks for the positive response from the reviewer**
> > >
> > > We sincerely appreciate your thoughtful input and appreciate the time you've taken to share your valuable feedback. We're pleased to have been able to address your concerns. Your insights and the ensuing discussion undoubtedly contribute to enhancing the quality of our work. Moreover, we're committed to integrating your suggestions and incorporating any new results into the upcoming version of our paper. Thank you once again for your contribution to our research.

---

### Author Rebuttal · Authors · 2023-08-09

In the author rebuttal PDF file, we include three additional figures for the reviewer's reference:

$\textbf{(1)}$ Fig.17: extended comparison with DiffEdit and Imagic for image editing;

$\textbf{(2)}$ Fig.18: ablation study of the Gradual Optimization for Token Updates;

$\textbf{(3)}$ Fig.19: progressively infusing the attention maps across diverse diffusion steps.

---

### Decision · Program_Chairs · 2023-09-21

**Decision:**

Accept (poster)

**Comment:**

This paper starts with experimental findings that the problem of unintended modifications of regions outside the targeted area proposes is caused by inaccurate cross-attention maps. dynamic prompt learning (DPL) to tackle in text-based image editing. This method employs leakage repairment losses to update the dynamic tokens for nouns, thereby forcing cross-attention maps to focus on correct nouns in the prompt.

Reviewers have identified a series of strengths and weaknesses of this paper as follows:

Pros:
1. The tackled problem is important.
2. The idea is interesting, simple yet effective. The method is well motivated and technically sound.
3. The paper is generally well written and easy to follow.
4. The effectiveness of the proposed method is demonstrated in extensive experiments.

Cons:
1. The method is not applicable when nouns do not correspond to intended areas, and is not applicable for real-time editing.
2. Lack of comparisons with important baselines.
3. Experiments are conducted on small-size datasets.
4. Lack of some ablation evaluations.

The authors have provided a detailed rebuttal, and most reviewers acknowledged that their major concerns are addressed. The AC agrees with the strengths and recommends accept, though would be preferable if necessary revisions could be provided according to reviewer’s comments.